# Open Eyes, Then Reason: Fine-grained Visual Mathematical Understanding in MLLMs

## Abstract

Current multimodal large language models (MLLMs) often underperform on mathematical problem-solving tasks that require fine-grained visual understanding. The limitation primarily arises from inadequate perception of geometric primitives during image-level contrastive pre-training (*e.g.*, CLIP). Current efforts to enhance MLLM performance have focused on scaling up mathematical visual instruction datasets and employing stronger LLM backbones, yet these approaches often neglect persistent visual recognition errors in MLLMs. In this paper, we systematically evaluate the visual grounding capabilities of state-of-the-art MLLMs and uncover a negative correlation between their visual grounding accuracy and problem-solving performance. Notably, even advanced models like GPT-4o demonstrate a significant error rate (70%) when identifying geometric entities, highlighting that fine-grained visual understanding remains a crucial bottleneck in visual mathematical reasoning. To address this, we propose a novel approach, *SVE-Math* (Selective Vision-Enhanced Mathematical MLLM), featuring a geometric-grounded vision encoder and a feature router that dynamically adjusts the contribution of hierarchical visual feature maps. Our model recognizes accurate visual primitives and generates precise visual prompts tailored to the language model's reasoning needs. In experiments, SVE-Math-Deepseek-7B outperforms other 7B models by 7.7% on MathVerse and is compatible with GPT-4V on MathVista. Despite being trained on smaller datasets, SVE-Math-7B matches the performance of models trained on significantly larger datasets, evaluated on GeoQA. Our findings provide critical insights for future research, highlighting the need for more effective integration of fine-grained visual understanding in MLLMs. We will release model weights, code, and instructions upon acceptance.

## 1 Introduction

Visual information plays a crucial role in mathematical problem-solving, where diagrams and visual representations are integral to understanding and reasoning. While Large Language Models (LLMs) have demonstrated impressive capabilities in textual mathematical reasoning (Yu et al., 2023; Ying et al., 2024; Azerbayev et al., 2023), their proficiency often diminishes when tasks require integrating visual data. The challenge intensifies when precise comprehension of geometric primitives—basic elements such as lines, circles, angles, boundaries, and junctions—is necessary to solve complex mathematical problems. Recent advancements in Multimodal Large Language Models (MLLMs) (Chen et al., 2022a; Liang et al., 2023; Kazemi et al., 2023; Gao et al., 2023a; Zhang et al., 2024b; Shi et al., 2024) have shown promise in addressing visual mathematical reasoning by incorporating both textual and visual inputs. These models typically rely on large-scale mathematical visual instruction datasets (Zhang et al., 2024b; Shi et al., 2024; Kazemi et al., 2023), which require MLLMs (OpenAI, 2023a;c; Su et al., 2023) to generate diverse descriptions for question-answer pairs involving geometric elements. While these approaches enhance the reasoning capabilities of MLLMs in the mathematical domain, they come with certain limitations. Constructing such datasets is time-consuming, labor-intensive, and requires substantial financial and human resources, often involving the use of advanced models like GPT-4o (OpenAI, 2023c) to generate diverse prompts for synthetic datasets.

Moreover, despite these efforts, even the most advanced MLLMs still exhibit notable shortcomings in accurately perceiving and grounding basic geometric primitives in mathematical diagrams. Our

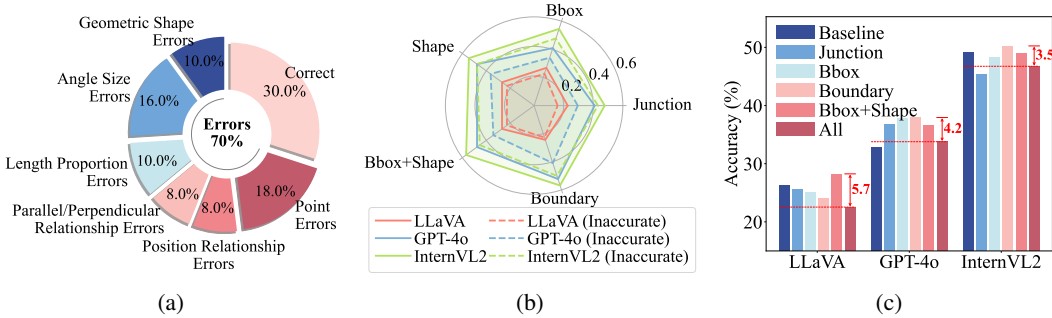

(a)                                   (b)                                   (c)

Figure 1: Analysis of MLLMs' performance in mathematical visual reasoning tasks from GeoQA test set. GPT-4o misperceived visual information in approximately 70% of cases involving geometric entities (Fig. 1a). Providing optimal geometric information enhances model performance, while redundant visual cues lower top-1 accuracy—even below the baseline achieved with only textual questions. (Fig. 1c). Model performance is sensitive to the accuracy of visual cues and a significant decrease ( 13.6%) in GPT-4o's top-1 accuracy is observed when provided with inaccurate bounding box locations and shape names (Bbox+Shape) (Fig. 1b).

systematic analysis reveals that visual recognition errors are prevalent and significantly impact the performance of MLLMs on mathematical reasoning tasks. We tasked LLMs with describing geometric entities in meticulously collected 100 images from the Geo170K dataset (Gao et al., 2023a), and then manually reviewed its responses to categorize the correct descriptions and error types. As demonstrated in Fig. 1a, we observed that GPT-4o misperceived visual information in approximately 70% of cases involving geometric entities. Correcting these visual perception errors led to a 12% overall accuracy improvement on corresponding mathematical questions (refer to Fig. 5a in the Appendix). This finding highlights that misunderstanding visual details remains a critical bottleneck in the mathematical reasoning capabilities of MLLMs.

To mitigate above challenges, we propose a novel approach termed SVE-Math (**S**elective **V**ision-**E**nhanced **Math**ematical MLLM) that diverges from the current trend of scaling up mathematical visual instruction datasets. Instead, we focus on enhancing the fine-grained visual perception capabilities of the model by training an auxiliary visual encoder, GeoGLIP (Geometric-Grounded Language-Image Pre-training), specifically tailored to recognize geometric primitives. Although existing mathematical datasets lack bounding box or pixel-level annotations, the training data generation process is simple yet highly efficient, *e.g.*, through the Matplotlib Python library. Moreover, training protocols for such visual-centric tasks are relatively straightforward compared to those for LLMs.

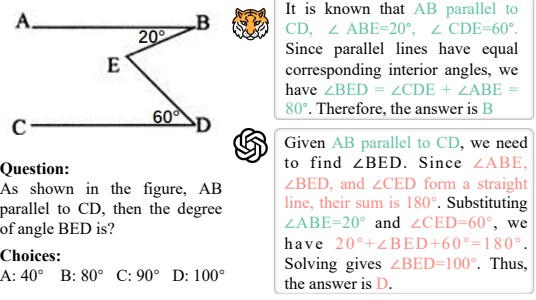

**Question:**
As shown in the figure, AB parallel to CD, then the degree of angle BED is?
**Choices:**
A: 40°  B: 80°  C: 90°  D: 100°

▷ GPT-4o struggles to accurately perceive mathematical elements, which impairs its ability to narrate their relationships for the reasoning process in LLMs. By integrating GeoGLIP, SVE-Math effectively grounds geometric elements and their positional relations (*e.g.*, ∠CDE), enabling accurate reasoning. See Appendix for more examples.

By incorporating GeoGLIP into existing MLLMs, we enable the models to *open their eyes* to the essential visual components of mathematical problems before engaging in reasoning.

Our hypothesis and design are inspired by observations as shown in Fig. 1b and Fig. 1c. Specifically, instructing MLLMs with fine-grained visual information, such as junction points and object locations, improves top-1 accuracy compared to providing only worded questions. However, providing all visual cues for solving a math question decreases accuracy, *e.g.*, a 4.2% decrease in GPT-4o's performance. These 'apples-to-apples' comparisons highlight that relevance is key—excessive information interferes with problem-solving (see § A.5 for a case study). Moreover, their performance is highly sensitive to the accuracy of visual cues. Providing inaccurate instructions, such as randomly generated box locations, significantly decreases performance. Given the inherent uncertainty in detecting geometric primitives by GeoGLIP, our initial approach utilizes global pyramid feature maps,

which capture information ranging from geometry-rich to semantic-rich representations. Their contributions are dynamically modulated by the feature router mechanism, resulting in the so-called visual soft prompts.

Our proposed SVE-Math has several key advantages. First, by enhancing the visual encoder to perceive geometric primitives, we directly tackle the root cause of geometrical visual recognition errors in mathematical reasoning tasks. Second, SVE-Math is efficient and practical, as it does not rely on the creation of large-scale instruction datasets or extensive human annotations. Third, our proposed auxiliary visual encoder and connector can be seamlessly integrated into any existing MLLM, enhancing its performance without modifying the reasoning components of language models.

We evaluate SVE-Math on several public mathematical benchmarks, and experimental results demonstrate its superior performance compared to models of the same or even larger sizes. Specifically, our model outperforms other 7B-parameter models and achieves comparable results to advanced 13B-parameter MLLMs, all while using a smaller-scale dataset for visual training (40K) and 60K + 110K for alignment and instruct learning, compared to the large 588K + 834K dataset used in MAVIS (Zhang et al., 2024b). These results highlight the effectiveness of our approach and underscore the importance of accurate visual perception in mathematical visual reasoning. In summary, our contributions are as follows:

- We systematically identify and analyze the impact of visual recognition errors on the mathematical reasoning performance of MLLMs, highlighting the critical role of accurately perceiving geometric primitives.
- We propose a novel method, SVE-Math, that enhances the visual perception capabilities of MLLMs by integrating a geometric-awareness visual encoder trained on small-scale box/pixel-level annotations, avoiding the need for large-scale instruction datasets.
- We design a connector mechanism featuring a feature router that effectively integrates the relevant geometric visual information into the language model, improving performance without altering the reasoning components.
- GeoGLIP integrates seamlessly with diverse LLM backbones without requiring modifications to their reasoning components. Extensive experiments demonstrate that SVE-Math outperforms existing models of comparable and larger sizes on mathematical benchmarks.

## 2 RELATED WORK

**Multimodal Large Language Models for Mathematics.** Large Language Models (LLMs) have recently garnered significant attention, with much research focused on text-based mathematical problem-solving, expanding mathematical datasets and utilizing data augmentation (Yu et al., 2023; Yue et al., 2023b; 2024; Luo et al., 2023). Meanwhile, advancements in vision-language alignment models, such as CLIP (Radford et al., 2021) and BLIP (Li et al., 2022a), have significantly progressed multimodal tasks, leading to the development of Multimodal Large Language Models (MLLMs) (Bai et al., 2023; Gemini Team, 2023; Ye et al., 2023a; Lin et al., 2023; Gao et al., 2024; Hu et al., 2024). With the rise of instruction-following LLMs, LLaVA (Liu et al., 2024b) adopts a linear layer to directly project visual tokens into LLMs, while MiniGPT-4 (Zhu et al., 2023) resamples visual tokens into fixed-length tokens, reducing the computation cost.

Building on these advancements, researchers have started to explore visual mathematical problem-solving using MLLMs. Unified frameworks like UniGeo (Chen et al., 2022a), UniMath (Liang et al., 2023), and GeomVerse (Kazemi et al., 2023) expand multimodal mathematical datasets and improve MLLM performance in geometry and diverse tasks. Leveraging current datasets, G-LLaVA (Gao et al., 2023a) constructed the Geo170K dataset, enhancing geometric problem-solving and surpassing GPT-4V (OpenAI, 2023c) on the MathVista benchmark (Lu et al., 2023) with only 7B parameters. GeoGPT4V (Cai et al., 2024a) further improved model performance on MathVista and Math-Vision (Wang et al., 2024) by creating a high-quality geometric problem dataset using GPT-4 and GPT-4V. MAVIS (Zhang et al., 2024b) specializes in mathematical tasks with a three-stage training pipeline including a math-specific vision encoder, while Math-LLaVA (Shi et al., 2024) introduced MathV360K, a large-scale dataset with high-quality images and diverse question-answer pairs to improve multimodal mathematical reasoning. These math-specific MLLMs have shown promising performance across several benchmark datasets (Yue et al., 2023a; Zhang et al., 2024a).

Figure 2: The diagram presents the architecture of SVE-Math, highlighting key innovations in the geometric-grounded vision encoder (GeoGLIP) and the feature router. Fine-grained visual understanding is achieved through a feature pyramid (attention maps displayed on the left), capturing hierarchical visual features ranging from geometry-rich to semantic-rich information. The feature router dynamically adjusts the contribution of these features to generate visual soft prompts. These prompts are then combined with CLIP visual tokens and textual inputs before being fed into the language model (LLM), enabling accurate visual perception and enhanced mathematical reasoning.

Despite these advancements, MLLMs still face challenges in multimodal mathematical tasks, particularly due to limitations in visual perception. While CLIP remains a common choice for many mathematical MLLMs and is known to benefit multimodal tasks, its limitations have also been identified. For instance, (Tong et al., 2024) examines 'CLIP-blind pairs', revealing that visually distinct images are often misinterpreted as similar, highlighting systematic shortcomings in CLIP's visual perception. These findings underscore the need for more specialized visual encoding methods tailored to mathematical contexts, as well as more rigorous evaluations of MLLMs' visual capabilities.

**Open-Set Object Detection.** Open-set object detection identifies arbitrary classes using existing bounding box annotations and language generalization. Methods like OV-DETR (Zareian et al., 2021), ViLD (Gu et al., 2022), DetCLIP (Yao et al., 2022), and Grounding DINO (Liu et al., 2024c) integrate language models with detection frameworks to improve category-specific detection. However, these models often struggle with small-scale object detection due to insufficient fine-grained visual understanding. GLIP (Li et al., 2022b) addresses this limitation by integrating textual information with visual region features early in the pipeline via a language-aware deep fusion mechanism, enhancing region-level embeddings. GLIP improves detection of smaller objects and demonstrates strong zero-shot capabilities. While GLIP's potential has been explored in various fields (Surís et al., 2023; Peng et al., 2023; Li et al., 2023), its application to mathematical reasoning, particularly in precise geometric entity description and fine-grained detail identification in mathematical diagrams, remains largely unexplored. Our work extends these concepts, developing a geometric-grounded language-image pre-training model (GeoGLIP) tailored for the unique demands of visual mathematical reasoning.

**Junction and Boundary Detection.** Junction and boundary detection are crucial in image processing and object recognition (Dollar et al., 2006; Maire et al., 2008; Parida et al., 1998), and can play a pivotal role in mathematical reasoning with geometric diagrams. Junctions represent points where lines intersect, and boundaries delineate object shapes. Traditional methods like Canny edge detection (Canny, 1986) and the Hough Transform (Duda & Hart, 1972) struggle with complex diagrams and fine-grained details required for accurate mathematical reasoning. Recent deep learning approaches, such as junction detection networks (Huang et al., 2018), detect key points by considering surrounding regions. Boundary detection models like Field of Junctions (FoJ) (Verbin & Zickler, 2021) use a bottom-up approach with 'generalized M-junctions' to detect contours and junctions.

## 3 METHODS

### 3.1 OVERVIEW

SVE-Math integrates visual understanding of geometric primitives with textual analysis to enhance the model's capability in solving mathematical problems involving visual elements. As illustrated in

Fig. 2, our pipeline builds upon the LLaVA-1.5 (Liu et al., 2023b) architecture (refer to §A.1), introducing key innovations in the GeoGLIP and visual feature connector. Feature maps from different layers of the GeoGLIP encoder are processed through the connector, where a feature router optimally integrates the feature pyramid into visual soft prompts by leveraging geometric information. These visual prompts are then fused with CLIP vision tokens, either along the sequence dimension or the channel dimension, and aligned with text embeddings via projection layers for visual understanding. Since channel-wise fusion offers better computational efficiency and comparable performance to sequence-based fusion in our experiments, we set channel-wise fusion as the default approach.

## 3.2 GEOMETRIC-GROUNDED LANGUAGE-IMAGE PRE-TRAININ

Our proposed GeoGLIP extends GLIP (Li et al., 2022b) to perform shape grounding, boundary and junction detection tasks with no human annotations. The architecture of GeoGLIP is shown in Fig. 7 of the Appendix. For shape grounding, we follow the same pipeline structure as the original GLIP model for bounding box detection (refer to §A.1 for pipeline details) but train it on the mathematical domain. Unlike the grounding task, which prioritizes semantic-rich visual information for localizing objects based on text inputs, boundary and junction detection require finer visual details. In general, feature pyramids encode information at different levels: higher-resolution features capture more geometric details, while lower-resolution features capture more semantic information. We employ a cross-resolution mixture to inject low-resolution features into high-resolution features, thereby improving visual understanding. Training details are provided in § A.6.1, and the training datasets are discussed in § A.3. Visualization results can be seen in Figures 9 and 10 of the Appendix.

**Boundary and junction detection.** GLIP-T utilizes Swin-Tiny as its backbone, producing a five-level feature pyramid $\{F_{\text{geo}}^i\}_{i\in\{1,2,3,4,5\}}$, where each level's resolution is progressively downscaled by a factor of 2. To enrich the high-resolution features with semantic information, we first pass the high-resolution tensor $F_{\text{geo}}^2$ (as the Query) and the low-resolution tensor $F_{\text{geo}}^4$ (as the Key and Value) to a Multi-Head Self Attention (MHSA) module. The resulting feature maps are upsampled by a factor of 2 and element-wise added to $F_{\text{geo}}^1$, producing $F_{\text{geo}}^{1^*}$. The rationale behind this design is to fully integrate the hierarchical object concepts at various scales produced by the downsampling layers with the high-resolution spatial information encoded by the initial embedding layer. Taking $F_{\text{geo}}^{1^*}$ as input, we then adopt two decoders for boundary and junction detection (see Fig. 8).

The boundary decoder consists of two successive perception blocks, each comprising an upsampling operation using nearest-neighbor interpolation, followed by a $3 \times 3$ convolution (Conv2d), batch normalization (BN2d), and ReLU activation. The final output is resized to the original image resolution using bilinear upsampling.

A junction represents the intersection of lines, determined by the intersection coordinates and the orientations of the lines. Accordingly, our junction decoder has two branches. The first branch estimates the confidence of a junction falling within each grid cell of the original image (using a $60 \times 60$ grid) and its relative position to the cell's center coordinates. The second branch predicts the orientations of the intersecting lines and their confidence in falling into one of 15 evenly spaced bins within each grid cell, where each bin covers 24 degrees, ensuring the full 360-degree range is divided evenly (15 bins $\times$ 24 degrees = 360 degrees). In the junction decoder, the input $F_{\text{geo}}^{1^*}$ is first processed through a perception block, where it is upsampled to a resolution of $60 \times 60$. Then, two separate Conv2D units predict the cell confidence and location, with output sizes of $60 \times 60 \times 1$ and $60 \times 60 \times 2$, respectively. Additionally, two other Conv2D units predict the bin confidence and orientation, both producing outputs of $60 \times 60 \times 15$. For further details, refer to training step 1 in §A.6.1 and the illustration in Fig. 8 of the Appendix.

## 3.3 CONNECTOR DESIGN

Recall our hypothesis that selecting key visual cues enhances mathematical visual problem-solving, while redundant information can hinder it. To manage the contribution of each feature and enhance the model's capacity, we propose a dynamic feature router $R$. The router $R$ is implemented as a simple Multi-Layer Perceptron (MLP) that takes as input the concatenation of the spatially averaged pooled feature maps from each level of GeoGLIP ($\bar{F}_{\text{geo}}^i \in \mathbb{R}^{1\times256}$) and the CLIP feature map ($\bar{F}_{\text{clip}} \in \mathbb{R}^{1\times1,024}$). It calculates the routing weights per feature ($\{\boldsymbol{w}^i\}_{i\in\{1,2,3,4\}} \in \mathbb{R}^{1\times4}$), functioning as a

**Image:** 1000 × 1000

| Shape | Coordinates |
|---|---|
| Circle | [185,142,593,593] |
| Rectangle | [116,56,311,441] |
| Text | [761,398,114,68], [142,382,148,77] |

Figure 3: Process for generating synthetic data with box- and pixel-level annotations, used to tranin our GeoGLIP visual encoder. 'Text' is a random string of alphanumeric characters with a length between 1 and 10, placed alongside other geometric objects, *i.e.*, circles and rectangles. Refer to Fig. 6 in the Appendix for the detailed flow chart.

soft router (Puigcerver et al., 2024). Alternative types of routers, such as sparse routers and constant routers, are also discussed in Sec. 4. The soft router's process is defined as:

$$\widehat{F}^i_{\text{geo}} = \mathbf{w}^i \cdot MLP \odot \mathcal{G} \odot F^i_{\text{geo}}, \quad \mathbf{w}^i = \sigma \odot R([\bar{F}^i_{\text{geo}}, \bar{F}_{\text{clip}}]), \tag{1}$$

where $F^i_{\text{geo}}$ is resized ($\mathcal{G}$) to match the spatial dimensions of $F_{\text{clip}}$ and processed by an MLP to align its channel dimensions. The scalar routing weights $\mathbf{w}^i$ are then applied to the respective features. The final $\widehat{F}_{\text{geo}}$ is generated either by element-wise addition of the weighted features $\widehat{F}_{\text{geo}} = \sum_{i=1}^{4} \widehat{F}^i_{\text{geo}}$, where the weights $\mathbf{w}^i$ are normalized using the SoftMax function (i.e., $\sum_{i=1}^{4} \mathbf{w}^i = 1$), or by channel-wise concatenation of the weighted features, where the weights are processed through a Sigmoid function, depending on the fusion strategy with $F_{\text{clip}}$.

Next, we explore strategies for fusing the soft prompts $\widehat{F}_{\text{geo}}$ with $F_{\text{clip}}$, either sequence-wise or channel-wise. In the sequence-wise method, additional tokens are added after the CLIP tokens, extending the sequence length. In contrast, channel-wise fusion combines all visual tokens along the channel dimension, maintaining the same sequence length. To enable the subsequent LLM to understand these visual components, the fused visual tokens are then fed into projection layers, which project the visual modality into the LLM's embedding space. Following the LLaVa-1.5 approach, we employ highly effective MLP projectors (linear layer + GELU + linear layer, a.k.a., `mlp2x_gelu`) for this task. In the sequence-wise approach, two separate projectors are applied for CLIP and soft prompts, respectively. For example, the projection matrices for the two linear layers, per projector, $\mathbf{\Phi_1}$ and $\mathbf{\Phi_2}$, have sizes of $1,024 \times 4,096$ and $4,096 \times 4,096$, where $4,096$ corresponds to the text embedding dimension. In the channel-wise approach, a single projector ($\mathbf{\Phi_1} \in \mathbb{R}^{5,120 \times 4,096}$ and $\mathbf{\Phi_2} \in \mathbb{R}^{4,096 \times 4,096}$) is used to process the combined visual tokens.

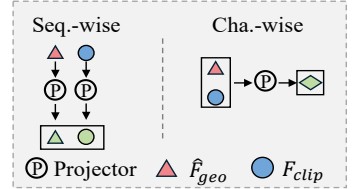

### 3.4 TRAINING SAMPLES FOR VISUAL-CENTRIC GEOGLIP

To enable GeoGLIP to perceive fine-grained mathematical elements, we supervise its training using datasets with box- and pixel-level annotations. The model is trained with a classical detection loss $\mathcal{L}_{det}$ (Eq. 2), a junction loss $\mathcal{L}_{junc}$ (Eq. 3), and a boundary loss $\mathcal{L}_{bodr}$ (the $\ell_2$ loss between predicted heatmap values and ground truth values). The detection loss $\mathcal{L}_{det}$ is applied to the shape grounding task, using synthetic images and FigureQA Kahou et al. (2018) training data annotated with bounding boxes and shape names (left panel of Fig. 3) . These annotations are stored in a COCO-style JSON file for seamless integration with standard GLIP. See §A.3 for details on the synthetic data engine and dataset statistics (Figures 5b and 5c).

For boundary and junction detection tasks, we leveraged off-the-shelf models (Huang et al., 2018; Verbin & Zickler, 2021) to extract junctions and boundaries as ground truth. In addition to our synthetic sampels, we incorporated the public dataset Geo170K Chen et al. (2021b) and generated the corresponding ground truth. Specifically, junction labels include intersection coordinates and line orientations. As noted, each grid cell and bin are responsible for predicting the coordinates and the orientations, and we have $60 \times 60$ cells&15 bins per cell. The labels are formatted as $JP_{ij} = (x_{ij}, c_{ij}, \{\theta_{ijk}, c^{\theta}_{ijk}\}_{k=1}^{K})$, where $x_{ij}$ denotes the junction center coordinates, $c_{ij} \in \{0, 1\}$ indicates the presence of a junction, $\theta_{ijk}$ is the angle of the $k$-th bin, and $c^{\theta}_{ijk} \in \{0, 1\}$ is the indicator for that bin (right panel of Fig. 3).

Table 1: **Results on testmini set of MathVerse** with the accuracy metric. The highest results for closed-source and open-source MLLMs are highlighted in red and blue respectively.

| Model | Base LLM | All | Text Dominant | Text Lite | Vision Intensive | Vision Dominant | Vision Only |
|---|---|---|---|---|---|---|---|
| | | Acc | Acc | Acc | Acc | Acc | Acc |
| *Baselines* | | | | | | | |
| Random Chance | - | 12.4 | 12.4 | 12.4 | 12.4 | 12.4 | 12.4 |
| Human | - | 67.7 | 71.2 | 70.9 | 61.4 | 68.3 | 66.7 |
| *LLMs* | | | | | | | |
| ChatGPT (Ouyang et al., 2022) | - | 26.1 | 33.3 | 18.9 | - | - | - |
| GPT-4 (OpenAI, 2023b) | - | 33.6 | 46.5 | 46.5 | - | - | - |
| *Closed-source MLLMs* | | | | | | | |
| Qwen-VL-Plus (Bai et al., 2023) | - | 11.8 | 15.7 | 11.1 | 9.0 | 13.0 | 10.0 |
| Gemini-Pro (Gemini Team, 2023) | - | 23.5 | 26.3 | 23.5 | 23.0 | 22.3 | 22.2 |
| Qwen-VL-Max (Bai et al., 2023) | - | 25.3 | 30.7 | 26.1 | 24.1 | 24.1 | 21.4 |
| GPT-4V (OpenAI, 2023c) | - | 39.4 | 54.7 | 41.4 | 34.9 | 34.4 | 31.6 |
| *Open-source MLLMs* | | | | | | | |
| LLaMA-Adapter V2 (Gao et al., 2023b) | LLaMA-7B (Touvron et al., 2023a) | 5.7 | 6.2 | 5.9 | 6.1 | 4.2 | 6.1 |
| ImageBind-LLM (Han et al., 2023) | LLaMA-7B | 9.2 | 11.4 | 11.3 | 8.9 | 11.2 | 3.4 |
| mPLUG-Owl2 (Ye et al., 2023b) | LLaMA-7B | 5.9 | 6.6 | 6.3 | 6.3 | 5.6 | 4.9 |
| SPHINX-Plus (Gao et al., 2024) | LLaMA2-13B | 12.2 | 13.9 | 11.6 | 11.6 | 13.5 | 10.4 |
| SPHINX-MoE (Gao et al., 2024) | Mixtral-8×7B (Jiang et al., 2024) | 15.0 | 22.2 | 16.4 | 14.8 | 12.6 | 9.1 |
| G-LLaVA (Gao et al., 2023a) | LLaMA2-7B | 16.6 | 20.9 | 20.7 | 17.2 | 14.6 | 9.4 |
| InternLM-XC2. (Dong et al., 2024) | InternLM2-7B (Cai et al., 2024b) | 16.5 | 22.3 | 17.0 | 15.7 | 16.4 | 11.0 |
| LLaVA-1.5 (Liu et al., 2023a) | Vicuna-13B | 7.6 | 8.8 | 7.6 | 7.4 | 7.4 | 6.9 |
| ShareGPT4V (Chen et al., 2023b) | Vicuna-13B | 13.1 | 16.2 | 16.2 | 15.5 | 13.8 | 3.7 |
| Math-LLaVA (Shi et al., 2024) | Vicuna-13B | 19.0 | 21.2 | 19.8 | 20.2 | 17.6 | 16.4 |
| LLaVA-NeXT (Li et al., 2024) | LLaMA3-8B (Team, 2024) | 19.3 | 24.9 | 20.9 | 20.8 | 16.1 | 13.8 |
| **SVE-Math-7B** | LLaMA2-7B | 21.2 | 26.4 | 23.2 | 22.9 | 18.0 | 15.4 |
| **SVE-Math-8B** | LLaMA3-8B | 23.4 | 29.3 | 23.4 | 23.1 | 21.1 | 20.3 |
| **SVE-Math-Deepseek-7B** | Deepseek-math-7B (Team, 2023) | 24.3 | 31.1 | 26.9 | 25.6 | 19.3 | 17.5 |

Table 2: **Results on testmini set of MathVista** with the accuracy metric. The highest results for closed-source and open-source MLLMs are highlighted. * means model trained on MathV360k.

| Model | Base LLM | All | FQA | GPS | MWP | TQA | VQA |
|---|---|---|---|---|---|---|---|
| | | Acc | Acc | Acc | Acc | Acc | Acc |
| *Baselines* | | | | | | | |
| Random Chance | - | 17.9 | 18.2 | 21.6 | 3.8 | 19.6 | 26.3 |
| Human | - | 60.3 | 59.7 | 48.4 | 73.0 | 63.2 | 55.9 |
| *Closed-source MLLMs* | | | | | | | |
| Qwen-VL-Plus (Bai et al., 2023) | - | 43.3 | 54.6 | 33.5 | 31.2 | 48.1 | 51.4 |
| GPT-4V (OpenAI, 2023c) | - | 49.9 | 43.1 | 50.5 | 57.5 | 65.2 | 38.0 |
| *Open-source MLLMs* | | | | | | | |
| mPLUG-Owl2 (Ye et al., 2023b) | LLaMA-7B | 22.2 | 22.7 | 23.6 | 10.2 | 27.2 | 27.9 |
| MiniGPT-v2 (Chen et al., 2023a) | LLaMA2-7B (Touvron et al., 2023b) | 23.1 | 18.6 | 26.0 | 13.4 | 30.4 | 30.2 |
| G-LLaVA (Gao et al., 2023a) | LLaMA2-7B | 25.1 | 19.1 | 48.7 | 3.6 | 25.0 | 28.7 |
| LLaVA-1.5 (Liu et al., 2023a) | Vicuna-13B | 27.7 | 23.8 | 22.7 | 18.9 | 43.0 | 30.2 |
| SPHINX-Plus (Gao et al., 2024) | LLaMA2-13B | 36.7 | 54.6 | 16.4 | 23.1 | 41.8 | 43.0 |
| **SVE-Math*-7B** | LLaMA2-7B | 37.4 | 31.9 | 53.9 | 29.0 | 41.4 | 30.8 |
| **SVE-Math*-Deepseek-7B** | Deepseek-math-7B (Team, 2023) | 48.7 | 37.6 | 62.0 | 48.1 | 48.1 | 35.8 |

# 4 EXPERIMENTS

## 4.1 EXPERIMENTAL SETUP

**Implementation Details.** Our work follows a structured three-stage training pipeline, including multi-task visual perception training for GeoGLIP, visual-language alignment, and mathematical instruction tuning for MLLMs (refer to §A.6.1 for details). We fine-tuned our GeoGLIP model using GLIP-T (Li et al., 2022b) as the pre-trained model, leveraging a combined dataset of 10,000 synthetic images, 20,672 images from FigureQA, and 9,426 images from the Geo170K training set. Training is conducted on 8 A100 GPUs with a batch size of 32. The base learning rate is set to $1 \times 10^{-5}$ for the language backbone and $1 \times 10^{-4}$ for all other parameters, and it is decreased by

a factor of 0.1 at 67% and 89% of the total training steps. We employ the same data augmentation strategies as GLIP, including random horizontal flipping and aspect ratio-preserving resizing with a minimum size of 800 pixels.

For multi-modal training, we freeze the GeoGLIP encoder. In Stage 2, we train only the projection layers to align diagram-language pairs. In Stage 3, we unfreeze both the projection layer and the LLM to perform comprehensive instruction-following tuning. We adopt LLaVA1.5-7B (Liu et al., 2023b) as the backbone of our MLLM, utilizing LLAMA-2 (Touvron et al., 2023b) as the language model and a pretrained vision transformer (CLIP ViT-L) (Radford et al., 2021) and our GeoGLIP as the visual encoders. Images are padded to squaresand resized to $448 \times 448$ pixels with a white background for processing by CLIP, and to $1000 \times 1000$ pixels for processing by GeoGLIP. We train SVE-Math for one epoch for cross-modal alignment and two epochs for instruction tuning on the Geo170K(Gao et al., 2023a) dataset, evaluating the model on GeoQA (Gao et al., 2023a) and the minitest set of MathVerse (Zhang et al., 2024a). To further enhance model performance and evaluate on MathVista (Lu et al., 2023), which encompasses a wider range of mathematical and visual tasks including IQTest, PaperQA, and IconQA, we incorporate the open-source MathV360k (Shi et al., 2024) dataset. We train our model on MathV360k using a batch size of 16 for one epoch with an initial learning rate of $3 \times 10^{-5}$.

**Evaluation Benchmarks.** We assess our SVE-Math using three well-established public mathematical benchmarks, MathVerse (Zhang et al., 2024a), GeoQA (Gao et al., 2023a), and MathVista (Lu et al., 2023)). MathVerse focuses on assessing multi-modal mathematical problem-solving with a combination of text and diagram-based reasoning tasks. GeoQA emphasizes geometric reasoning, where the model must interpret geometric shapes and solve related questions. MathVista includes a diverse set of mathematical and visual tasks, providing a comprehensive evaluation across various reasoning and problem-solving domains.

**Evaluation Metrics.** We adopt top-1 accuracy to evaluate our model on these benchmarks. Our evaluation process follows the protocols defined by the respective datasets, where LLMs are used to extract predicted answers from the model's responses. Accuracy is determined by comparing these predicted answers against the corresponding ground truths.

## 4.2 MAIN RESULTS

Table 1 presents the comparison results on the testmini set of MathVerse, where SVE-Math-7B outperforms all models using LLaMA2-7B as the base LLM by a significant margin (a 5.5% increase) and achieves comparable top-1 accuracy to the most powerful open-source LLaVA-NeXT (Liu et al., 2024a) with 8B size (19.3% *vs*. 21.2%). When using DeepSeek-Math-7B-Instruct Team (2023) as the base LLM, our model's performance further increases by an additional +3.1%. Notably, even on the challenging MathVista benchmark, our model outperforms the advanced SPHINX-Plus-13B (Gao et al., 2024), and is compatible with close-sourced GPT-4V OpenAI (2023c), as shown in Table 2. This superior performance underscores the importance of fine-grained visual perception in enhancing the mathematical reasoning capabilities of MLLMs.

Tables 3 and 4 present our model's performance on plane geometry and function analysis tasks, respectively. Compared to the second-best model, MAVIS (Zhang et al., 2024b), which is trained on an $8\times$ larger mathematical visual instruction dataset, SVE-Math with LLaMA2-7B as LLM demonstrates better reasoning and generalization capabilities. Constructing large instruction datasets for training MLLMs is labor-intensive and costly, whereas synthetic datasets for training traditional visual-only tasks offer a more efficient solution. This positions our method as a promising alternative and orthogonal direction for mathematical visual reasoning tasks.

Notably, the effectiveness of geomatic soft visual prompts is evidenced by comparison SVE-Math-7B with G-LLaVA in Tables 1-3. This comparison, conducted under controlled conditions, ensures that both G-LLaVA and our model utilize the same LLM backbone (LLaMA2-7B) and the instruction training dataset, with +7.7% on MathVerse +12.3% on MathVista and +2.8 % on GeoQA.

## 4.3 ABLATION ANALYSIS

**Effect of cross-resolution mixture.** We designed four additional variants to demonstrate the effectiveness of our cross-resolution mixture approach. Recall that we have five feature lev-

Table 3: Comparison of geometric numerical answer accuracies (%) on **GeoQA**.

| Model | Accuracy (%) |
|---|---|
| Random Chance | 25.0 |
| Frequent Guesses | 32.1 |
| *Top-10 Accuracy* | |
| NGS (Chen et al., 2021a) | 56.9 |
| DPE-GPS (Cao & Xiao, 2022) | 62.7 |
| SCA-GPS (Ning et al., 2023) | 64.1 |
| *Top-1 Accuracy* | |
| Geoformer (Chen et al., 2022b) | 46.8 |
| UniMath (Liang et al., 2023) | 50.0 |
| G-LLaVA (Gao et al., 2023a) | 64.2 |
| MAVIS-7B (Zhang et al., 2024b) | 66.7 |
| **SVE-Math-7B** | 67.0 |
| **SVE-Math-Deepseek-7B** | 72.8 |

Table 4: Comparison of model performance on **FunctionQA of MathVista.**

| Model | Accuracy (%) |
|---|---|
| Random Chance | 22.5 |
| *Closed-source MLLMs* | |
| CoT GPT-4 (OpenAI, 2023b) | 35.0 |
| PoT GPT-4 (OpenAI, 2023b) | 37.0 |
| Multimodal Bard (Google, 2023) | 45.5 |
| GPT-4V (OpenAI, 2023c) | 69.5 |
| *Open-source MLLMs* | |
| LLaVA (Liu et al., 2023b) | 20.5 |
| LLaMA-Adapter V2 (Gao et al., 2023b) | 32.0 |
| LLaVA-NeXT (Liu et al., 2024a) | 33.7 |
| SPHINX-MoE (Gao et al., 2024) | 34.6 |
| MAVIS-7B (Zhang et al., 2024b) | 40.3 |
| **SVE-Math-7B** | 40.5 |
| **SVE-Math-Deepseek-7B** | 45.1 |

els $\{F_{geo}^i\}_{i \in \{1,2,3,4,5\}}$ with different resolutions, each with different resolutions, ranging from geometric-rich to semantic-rich information. The cross-resolution mixture aims to generate the input $F_{geo}^{1*}$ for the boundary and junction decoders, with the expectation that $F_{geo}^{1*}$ captures more informative visual information to benefit boundary and junction detection tasks.

Using boundary detection as an example, we first used the semantic-rich $F_{geo}^5$ as input to the boundary decoder. As shown in Fig. 4a, the decoder fails to generate clear boundaries, resulting in a blurred output. Next, we used the geometric-rich $F_{geo}^1$, which performs better (Fig. 4b), showing some visible boundaries. To further enhance the results, we applied a cross-resolution attention mechanism (classic Multi-Head Self-Attention, MHSA) between $F_{geo}^2$ and $F_{geo}^4$, improving boundary detection as seen in Fig. 4d. Since boundary detection benefits from geometric-rich information, we upsampled the cross-correlated features by a factor of 2 and added them element-wise with $F_{geo}^1$, producing the best visualization results, especially for finer details (Fig. 4e). Finally, to assess the importance of cross-resolution attention, we replaced it with element-wise addition. As expected, the boundaries became blurred (Fig. 4c) due to the reduced receptive field. Replacing addition with the attention mechanism yields similar boundary results but decreases object detection mAP from 95.3% to 92.4% on our synthetic test set. Therefore, our mixture process integrates both cross-resolution attention and addition operations.

**Key Factors in Connectors.** Our connector bridges the soft visual prompts $\widehat{F}_{geo}$ with the CLIP visual tokens $F_{CLIP}$ using either channel-wise or sequence-wise fusion methods. We examine two key factors: the inclusion of all visual cues and the use of soft routing. Additionally, for sequence fusion, we explore varying feature resolution sizes. All ablations are conducted on the GeoQA test set. The summary is presented in Fig. 5b, with detailed top-1 accuracy listed in Fig. 5c. Specifically, for smaller resolutions, we resize the pyramid features from GeoGLIP to lengths of 15%, 20%, 25%, and 40% of the length of $F_{CLIP}$, respectively, and then sequentially append them to $F_{CLIP}$.

Next, we examine the impact of the number of projection experts. The default channel concatenation setup utilizes a single expert with a `mlp2x_gelu`. In the multi-expert ablation, where two sequential `mlp2x_gelu` are applied, the top-1 accuracy drops from 66.98% to 64.32% (-2.66%), as shown in Fig.5c. For sequence-wise fusion, which uses two separate projectors by default, we ablate shared parameters across these projectors, making them act as a single-projection expert. Fig. 5c shows that the multi-expert setup enhances sequence-wise performance compared to shared parameters (a.k.a., a single expert), boosting accuracy from 64.32% to 66.58% (+2.26%). We hypothesize that the improvement in sequence-wise fusion may stem from the added flexibility in handling heterogeneous inputs, whereas in channel-wise fusion, it could introduce unnecessary complexity and redundancy.

**Feature router types and impact of individual feature maps in GeoGLIP.** We examine three types of routers: constant, sparse, and the default soft router $R$. The constant router assigns equal weights $w^i = 0.25$ to each $F_{geo}^i$, while the sparse router selects only one feature map of GeoGLIP with $w^i \in \{0, 1\}$. As expected, in the sparse router, $F_{geo}^{1*}$ with more geometric information, achieves the highest accuracy. As shown in Table 5a, the soft router outperforms the others, demonstrating its effectiveness for dynamic routing of multiple signals.

Figure 4: Qualitative boundary visualization results. Semantic-rich features with the lowest resolution lead to blurred boundaries (Fig. 4a), while geometric-rich features with the highest resolution improve clarity (Fig. 4b). The cross-resolution mixture yields the best results (Fig. 4e), compared with using either element-wise addition (Fig. 4c) or MHSA alone (Fig. 4d). Zoom in for best view.

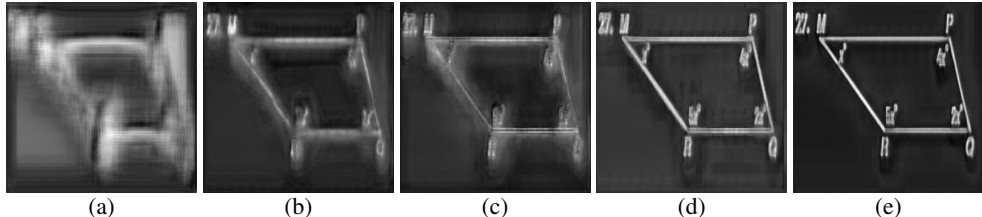

|  | (a) | (b) | (c) | (d) | (e) |

Table 5: Ablation results w.r.t. top-1 accuracy on GeoQA. Tab. 5a shows results for feature router types; Fig. 5b highlights key factors for connector designs, with detailed accuracy in Fig. 5c.

|  | Seq.-wise | Cha.-wise |
|---|---|---|
| Constant $R$ | 63.9 | 62.8 |
| Sparse $R$ | $F_{geo}^{1^*} \to$ 64.2 | $\to$ 64.9 |
|  | $F_{geo}^{3} \to$ 61.1 | $\to$ 61.8 |
|  | $F_{geo}^{4} \to$ 61.9 | $\to$ 62.3 |
|  | $F_{geo}^{5} \to$ 61.9 | $\to$ 61.6 |
| Soft $R$ | **66.6** | **67.0** |

(a)

(b)

(c)

**Necessity of CLIP.** While GeoGLIP provides rich geometric visual features, the general visual features provided by models such as CLIP are also crucial. We designed a variant that excludes the CLIP visual encoder, relying solely on our soft prompts from the GeoGLIP visual encoder. Accuracy dropped from 66.6% to 64.7% for sequence fusion and from 67.0% to 65.3% for channel fusion. These results demonstrate that while CLIP may not perceive fine-grained visual details, its general visual features still benefit text-visual alignment in MLLM training, making such models indispensable in multi-modal mathematical reasoning.

**Imapct of math-specific fine-tuning for GeoGLIP.** We utilized the original hierarchical pyramid features from the GLIP visual encoder. To ensure a fair comparison, we utilize the same resolution feature maps: the first layer with the largest resolution and the last three layers with smaller resolutions. This resulted in a drop from 67.0% to 65.3%, with only a minimal +1.1% improvement over G-LLaVA. The slight improvement likely stems from integrating high-resolution vision features, which are not sensitive to geometric details, as GLIP fails to detect basic geometric shapes (Fig. 9).

## 5 CONCLUSION

In this paper, we mitigate the limitations of current mathematical MLLMs by identifying the significant bottleneck caused by their inability to accurately perceive geometric primitives, which are crucial for mathematical reasoning involving visual elements. We proposed SVE-Math, a novel vision-centric approach that enhances mathematical visual reasoning by integrating a geometric-awareness visual encoder trained through multi-task objectives such as shape detection, junction detection, and boundary detection. Our method avoids the labor-intensive process of building large-scale mathematical visual instruction datasets, offering a more efficient and practical solution. By designing a feature router that dynamically adjusts the contribution of each visual cue, we generate soft prompts that guide the language model toward better mathematical reasoning without overwhelming it with redundant or irrelevant visual data. Extensive experiments across three public mathematical benchmarks demonstrate the effectiveness of SVE-Math, as SVE-Math outperforms similarly sized 7B-parameter models and achieves comparable results to advanced 13B-parameter MLLMs, despite being trained on smaller datasets. We believe our work introduces a new perspective on solving mathematical problems in a visual context, emphasizing the critical role of fine-grained visual grounding and adaptive visual cueing mechanisms.

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
