OPEN EYES, THEN REASON: FINE-GRAINED VISUAL MATHEMATICAL
UNDERSTANDING IN MLLMS

# A APPENDIX

In this supplementary material, we illustrate the related background for our method (§ A.1), provide a detailed description for GeoGLIP (Geometric-Grounded Language-Image Pre-training) pipeline (§ A.2), explain the process of synthetic data generation, and outline the datasets used for training GeoGLIP (§ A.3), present visualizations of the GeoGLIP detection results (§ A.4), offer case studies that illustrate the practical application of our feature router mechanism and chain-of-thought (CoT) reasoning results (§ A.5), demonstrate the training details/efficiency of SVE-Math (§ A.6) and examine our model's limitations while outlining potential directions for future work (§ A.7).

## A.1 BACKGROUND

**Grounded Language-Image Pre-training (GLIP).** GLIP (Li et al., 2022b) unifies detection and grounding by reformulating object detection as phrase grounding. It accepts paired image-text inputs, where the text consists of candidate detection categories, such as the 80 COCO object class names joined by '.', *i.e.*, person. bicycle. car. $\cdots$ toothbrush. In GLIP, object classification logits in the box classifier (traditional object detection) are replaced with word-region alignment scores, computed as the dot product between region visual features and phrase language features. GLIP operates as a two-stage detector, composed of: 1) A Swin Transformer as a visual encoder, which extracts features $F_I$ of images $X_I$ and passes $F_I$ to a Region Proposal Network (RPN) to generate region coordinates, and then corresponding region features $O_I$ are cropped from $F_I$; 2) A pre-trained BERT model as the language encoder, to embed the input text $X_L$ into token embeddings $P_L$; 3) A language-aware deep fusion module $\text{Fus}_{IL}$ that fuses $O_I$ and $P_L$ in the last few encoding layers. The final alignment scores $S_{\text{ground}}$, calculated as:

$$O_I = \text{RPN}(\text{Swin}(X_I)), \quad P_L = \text{BERT}(X_L), \quad O_I', P_L' = \text{Fus}_{IL}(O_I, P_L) \quad S_{\text{ground}} = O_I', {P_L'}^{\top}.$$

**Large Language and Vision Assistant (LLaVA).** We adopt (Large Language and Vision Assistant) LLaVA's architecture (Liu et al., 2023b) as the basis. LLaVA leverages the complementary strengths of pre-trained large language models and visual encoders to perform multi-modal tasks, consisting of a large language model $f_\phi$ (Vicuna (Chiang et al., 2023)), a vision encoder (CLIP, ViT-L/14) (Radford et al., 2021), and a projection layer. The projection layer projects the visual embedding from the vision encoder into the text embedding space. LLaVA begins by processing an input image $X_I$ through the CLIP visual encoder, which extracts visual features $F_I = \text{CLIP}(X_I)$. To bridge the gap between the image features and the language model's word embedding space, LLaVA applies a simple linear projection matrix $\mathbf{\Phi}$, converting visual features $F_I$ into visual tokens $H_I$, which are compatible with the language embedding space:

$$H_I = \mathbf{\Phi} \cdot F_I, \text{ with } F_I = \text{CLIP}(X_I)$$

The visual tokens $H_I$ and language instruction tokens $P_L$ are passed into the language model for joint reasoning and language generation as $f_\phi([H_I, P_L])$.

## A.2 GEOGLIP

The GeoGLIP pipeline is shown in Fig. 7, where the RPN and language-aware deep fusion details are omitted for clarity. The GeoGLIP takes image-text paired as input: an image containing geometric shapes and a text listing the shape classes (*i.e.*, 'circle. trapezoid. triangle. ... line.'). These inputs are processed by the GeoGLIP encoder, which generates feature pyramids at multiple scales ($F_{\text{geo}}^1, F_{\text{geo}}^2, F_{\text{geo}}^3, F_{\text{geo}}^4, F_{\text{geo}}^5$). Each feature pyramid contains different levels of detail, capturing varying levels of geometric information. These features are then routed to three separate detectors: 1) Shape Detector: identifies and localizes basic geometric shapes by generating bounding boxes for objects within the image; 2) Junction Detector: detects junctions or intersections of geometric entities in the image; 3) Boundary Detector: identifies boundaries of geometric shapes, refining their outlines for more accurate representation. The combination of the feature pyramids with task-specific detectors allows GeoGLIP to perform fine-grained visual tasks in a mathematical context.

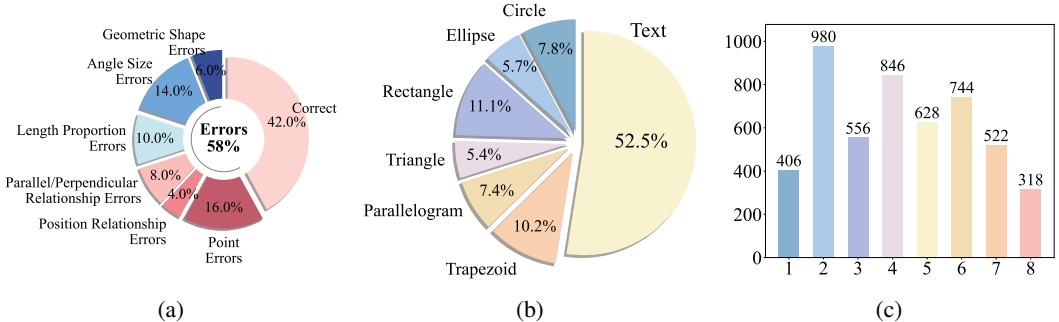

(a)  (b)  (c)

Figure 5: Fig. 5a presents the statistics of top-1 accuracy after manually correcting the visual perception errors shown in Fig.1a of the main paper, which initially caused incorrect answers to mathematical questions. Specifically, we restated the output of GPT-4o w.r.t. each type of visual recognition error and calculated the accuracy of its answers. Overall, correcting these visual perception errors led to an approximate 12% increase in accuracy on the corresponding mathematical questions. Fig. 5b and Fig. 5c present the data statistics for synthetic math-specific datasets, including the distribution of geometric shapes/classes and the number of objects per image. Each geometric object has a 70% probability of being assigned an alphanumeric text, leading to a higher proportion of the 'Text' class.

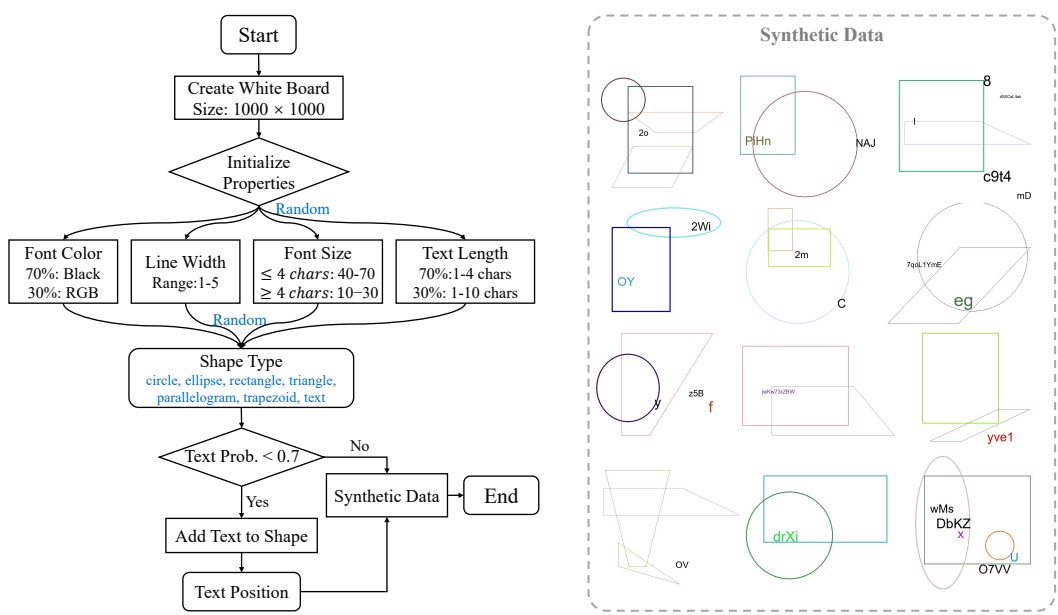

Figure 6: The flow diagram depicts the process for generating synthetic math-specific datasets, along with visualizations of the generated data samples.

In Fig. 9, we illustrate detailed designs about junction and boundary detectors:

- Junction Detector: The detector processes the feature $F_{geo}^{1^*}$ through a decoder, identifying the confidence of junction points within each grid cell and their relative positions. It also predicts the orientations and confidence levels of intersecting lines within the grid, split into multiple angular bins to cover the 360-degree range.

- Boundary Detector: It employs two successive perception blocks and upsampling operations to restore the feature map to the original image resolution for boundary decoding.

Both detectors use multi-resolution feature maps from the GeoGLIP encoder, and specific design for each task is optimized to capture relevant geometric properties, contributing to enhanced mathematical visual reasoning. Refer to § 3.2 of main paper for more details.

### A.3 Training Dataset for GeoGLIP

Notably, our synthetic math-specific datasets diffies from the traditional mathematical instruction datasets, and we do not create or use any additional self-generated instruction datasets beyond the publicly available Geo170K Gao et al. (2023a) and MathV360K Shi et al. (2024) datasets for MLLM training. Instead, our synthetic samples, annotated with box/pixel-level details, are exclusively utilized to train the GeoGLIP. Compared to constructing mathematical instruction datasets, our synthetic data generation process is significantly more efficient and resource-friendly. It does not require manual labeling, as all data can be programmatically generated, *e.g.*, through the Matplotlib Python library. In contrast, constructing instruction datasets often relies on GPT-4o to create diverse prompts and necessitates human intervention, making the process labor-intensive and costly.

**Shape grounding.** To generate *synthetic datasets* for object grounding tasks, we employ an automated Python-based approach that efficiently creates images containing geometric shapes and text with associated bounding boxes, class labels, and annotations. The geometric categories include shapes like circles, ellipses, rectangles, triangles, parallelograms, trapezoids, and text. A variable number of basic geometric shapes and alphanumeric text elements are generated, with font sizes dynamically adjusted according to text length. These shapes are randomly distributed within a $1000 \times 1000$ pixel canvas, while text is positioned either inside or adjacent to the shapes with a 70% probability. Bounding boxes are then calculated for each shape and text element, ensuring they remain within image bounds. Finally, shapes and text are assigned class labels and coordinates, saved in a COCO-style JSON file for seamless integration with standard GLIP. Fig. 6 shows the detailed flow diagram. Fig. 5b and Fig. Fig. 5c present the data statistics for synthetic math-specific datasets, including the distribution of geometric shapes and the number of objects per image. Additionally, we incorporated 20,672 images from the *FigureQA* training dataset with bounding box annotations for the shape grounding task.

**Junction and boundary detection.** We utilized off-the-shelf models (Huang et al., 2018; Verbin & Zickler, 2021) to extract junctions and boundary as ground truth on both our *synthetic dataset* and public *Geo170K* training images. We then designed junction and boundary heads, parallel to the object detection head, with all tasks sharing the same visual encoder. Through this multi-task learning approach, our GeoGLIP can perceive rich visual information in the mathematical domain.

### A.4 GeoGLIP Detection visualizations

Fig. 9 illustrates shape detection results on Geo170K, FigureQA and our synthetic test dataset, while Fig. 10 presents the results for boundary and junction detection. Our detector successfully localizes basic geometric shapes and junction points while providing pixel-level boundary results in most cases. However, in complex scenarios such as overcrowded or occluded settings, the detector may struggle. Moreover, in junction detection, some failure cases involve numerous detections but with low accuracy. This issue arises due to noisy ground truth during the training phase, as manually labeling junctions is tedious and time-consuming. To address this, we use an off-the-shelf model (Huang et al., 2018) to generate ground-truth labels for junction detection. However, since this model was trained on images of man-made environments, it faces an out-of-domain challenge when applied to geometric objects, resulting in labels that are not fully accurate. Improving the accuracy of these labels would significantly enhance junction detection performance.

### A.5 Case studies

**Selective visual information helps reasoning.** Fig. 11 showcases GPT-4o's responses based on additional visual information from geometric primitives, alongside the question, choices, and diagram ⟨image⟩ as inputs. We provide hard-coded coordinates for bounding boxes and junctions using instructions such as: "there is a bounding box at ⟨x, y, w, h⟩ (the normalized center point and width/height)" with shape names ⟨geometric shape⟩ (if shape information is provided), or "candidate junction point ⟨x, y⟩. For boundary information, we use "⟨boundary image⟩ is the boundary sketch related to the main diagram" as instructions. The right side visualizes the provided visual cues in the original geometric diagram for clarity, though these images are not input into GPT-4o. Fig. 11 highlights the importance of providing relevant visual prompts for each case; otherwise, redundant information may interfere with the solving process. For example, in case 1, bounding box coordinates per object can be distracting when solving a perimeter question compared to junction lo-

cations. In contrast, pixel-level visual information (boundary) aids the model in perceiving complex geometric shapes, such as polygons and circles, and is beneficial for calculating overlap regions, while relying on junctions may lead to biased answers. In practice, selecting supporting information for each case is labor-intensive and requires the involvement of math experts. We address this challenge by using the feature router, which automatically learns which fine-grained visual information is important during the training stage.

Notably, we do not claim that the feature router can explicitly select specific types of visual information, such as bounding boxes, junctions, or shapes. This is because the inputs to the feature pyramid of the GeoGLIP visual encoder do not clearly represent each type of information in a distinct manner. Since GeoGLIP is trained on multiple tasks using a shared visual encoder, it becomes challenging to determine which specific feature maps correspond to which an individual learning task. What our findings emphasize is the importance of selecting optimal visual cues, demonstrating that while accuracy is crucial, more information does not always lead to better performance—relevance is key. We anticipate that more advanced selection techniques could further enhance mathematical problem-solving in visual contexts. Refer to Sec. A.7 for our future research directions.

**Response comparison.** Fig. 12 presents case studies comparing our SVE-Math-Deepseek-7B with GPT-4o on the MathVerse testmini set. These examples highlight the strengths of SVE-Math-Deepseek-7B in providing precise geometric visual information, enabling clear and logically grounded mathematical reasoning in its responses. For instance, our model demonstrates sensitivity to the positions of individual points/junctions, effectively capturing the relationships between different lines. As shown in Fig. 12a, it successfully identifies angle 1 and its relationship with angle BEF, enabling correct reasoning and answers. In contrast, GPT-4o fails to recognize these relationships, leading to flawed reasoning and incorrect answers.

Fig. 13 and Fig. 14 present a Chain-of-Thought (CoT) comparison among SVE-Math-Deepseek-7B, GPT-4V, and InternVL2. The results clearly demonstrate that providing geometry-aware visual cues significantly aids LLMs in understanding the relationships between geometric elements, thereby enhancing the entire reasoning process. In contrast, the other two MLLMs fail to achieve this level of understanding, leading to incorrect reasoning and outcomes. This demonstrates that without accurately recognizing visual elements, even strong LLMs struggle with reasoning tasks. As shown in GPT-4V's output, its initial misidentification of mathematical elements results in an incorrect Chain-of-Thought (CoT) response.

## A.6 MATHEMATICAL VISUAL TRAINING AND EFFICIENCY

### A.6.1 TRAINING DETAILS

Our work follows a structured three-stage training pipeline, including multi-task visual perception training for GeoGLIP, visual-language alignment, and mathematical instruction tuning for MLLMs.

**Stage 1:** To enable the visual encoder in GeoGLIP to ground geometric entities in mathematical diagrams, we utilize synthetic and FigureQA training images annotated with bounding boxes for the *grounded pre-training*. Specifically, we fine-tune a pre-trained GLIP-T model (with Swin-Tiny as the backbone), adhering to the GLIP detection loss defined as:

$$\mathcal{L}_{det} = \mathcal{L}_{rpn} + \mathcal{L}_{cls} + \mathcal{L}_{reg} \tag{2}$$

where $\mathcal{L}_{rpn}$ refines the region proposals generated by the RPN, $\mathcal{L}_{cls}$ applies binary sigmoid loss to alignment scores, and $\mathcal{L}_{reg}$ uses smooth $\ell_1$ loss for bounding box regression.

Following the process in (Huang et al., 2018), for the *junction detection* task, the input image is divided into mesh grids, with each grid cell responsible for detecting a junction if its center falls within the cell. Each $ij$-th cell predicts a confidence score $c_{ij}$, indicating the likelihood of a junction in that cell. Since a junction represents the intersection of lines, the number of predictions per cell varies depending on the number of lines intersecting. To capture orientations, each cell is further divided into $K$ equal bins (default $K = 15$), with each bin spanning 24 degrees to cover the full 360-degree range. Each junction is represented as $JP_{ij} = (x_{ij}, c_{ij}, \{\theta_{ijk}, c_{ijk}^{\theta}\}_{k=1}^{K})$, where $x_{ij}$ denotes the junction center coordinates, $c_{ij} \in [0, 1]$ is the confidence score for the presence of a junction, $\theta_{ijk}$ is the angle of the $k$-th bin, and $c_{ijk}^{\theta}$ is the confidence score for that bin.

The loss function for junction detection consists of four terms. Given a set of ground truth junctions $JP = jp_1, \ldots, jp_N$ in an image, the loss function is formulated as:

$$\mathcal{L}_{junc} = \lambda_{loc} \cdot (\mathcal{L}_{loc}^c + \mathcal{L}_{loc}^b) + \lambda_{conf} \cdot (\mathcal{L}_{conf}^b + \mathcal{L}_{conf}^b). \tag{3}$$

The default values for the weights in Eq. 3 are $\lambda_{loc} = 0.1$ and $\lambda_{conf} = 1$, where the superscripts $c$ and $b$ refer to cell and bin, respectively. Specifically, we apply the binary cross-entropy loss for both $\mathcal{L}_{conf}^c$ and $\mathcal{L}_{conf}^b$, and use $\ell_2$ loss to measure the relative position of the predictions against the ground truth for $\mathcal{L}_{loc}^c$ and $\mathcal{L}_{loc}^b$. Refer to (Huang et al., 2018) for more details. In the *boundary detection* task, $\mathcal{L}_{bodr}$ minimizes the $\ell_2$ loss between the estimated heatmap values and the ground truth values.

Our final loss function for multi-task visual perception training is defined as:

$$\mathcal{L}_{\text{vis}} = \mathcal{L}_{det} + \mathcal{L}_{junc} + 5 \cdot \mathcal{L}_{bodr}, \tag{4}$$

where the weight for $\mathcal{L}_{\text{bodr}}$ is set to 5, while the weights for $\mathcal{L}_{\text{det}}$ and $\mathcal{L}_{\text{junc}}$ are kept at 1.

**Stage 2 & 3:** During both phases, we freeze the GeoGLIP encoder. In Stage 2, we train only the projection layers to align diagram-language pairs. In Stage 3, we unfreeze both the projection layer and the LLM to perform comprehensive instruction-following tuning, culminating in SVE-Math-7B. For these two stages, we employ the conventional LLaVA loss, formulated as:

$$\mathcal{L}_{llm} = -\sum_{t=1}^{L} \log p \left[ S_{tar}^t | f_\phi(s_{tar}^{(<t)}, S_{in}, I) \right], \tag{5}$$

where $f_\phi$ denotes the model parameterized by $\phi$, $I$ corresponds to the figure, $S_{tar}$ and $S_{in}$ represent the target and input sentences, respectively; $S_{tar}^t$ refers to the $t$-th token of the target output, and $L$ denotes the sequence length.

### A.6.2 Efficiency

SVE-Math-7B introduces minimal computational overhead, as detailed in the below comparison Table 6. The GeoGLIP and Connector contribute an additional parameter size of 32.65MB and 8.73MB, and the Projectors accounting for 16.13MB. The inference time per sample increases slightly, from 19.80s to 20.04s (+0.24s). Training is conducted on 8 A100 GPUs with a batch size of 128 using the Math360K dataset, which includes 40K images and 360K question-answer pairs. The total training time shows only a marginal increase, from 10.35h to 10.54h (+0.19h), demonstrating scalability for larger models and datasets.

Table 6: Comparison of computational overhead and parameter size for G-LLaVA and SVE-Math.

| #Parameter size | GeoGLIP | Connector | Projectors | Time (inference/sample) | Time (training/Math360K) |
|---|---|---|---|---|---|
| G-LLaVA | - | - | 16.52MB | 19.80s | 10.35h |
| **SVE-Math** | 32.65MB | 8.73MB | 31.20MB | 20.04s | 10.54h |

### A.7 Limitations and Further research

Our research aims to offer a new perspective on solving mathematical visual reasoning problems by first training a vision-centric model to provide visual prompts for LLMs, rather than focusing on creating large visual instruction fine-tuning datasets for MLLMs. Despite the effectiveness of our approach, there are several limitations to consider. First, the reliance on synthetic data for visual tasks may not fully capture the complexity of real-world geometric problems, potentially limiting generalization to more diverse datasets. Additionally, while the feature router provides automatic selection of relevant visual cues, it may not always perfectly align with human intuition or domain-specific knowledge, particularly in cases requiring more intricate reasoning.

For future research, one promising direction is to extend our method by incorporating real-world mathematical datasets to improve generalization and robustness. However, this will require some human labeling processes, as existing mathematical datasets lack detailed box or pixel-level annotations. Incorporating such annotations would provide a more accurate and fine-grained understanding of visual elements in mathematical problems, allowing models to better generalize to real-world scenarios. Developing efficient semi-automated labeling techniques or combining expert annotations

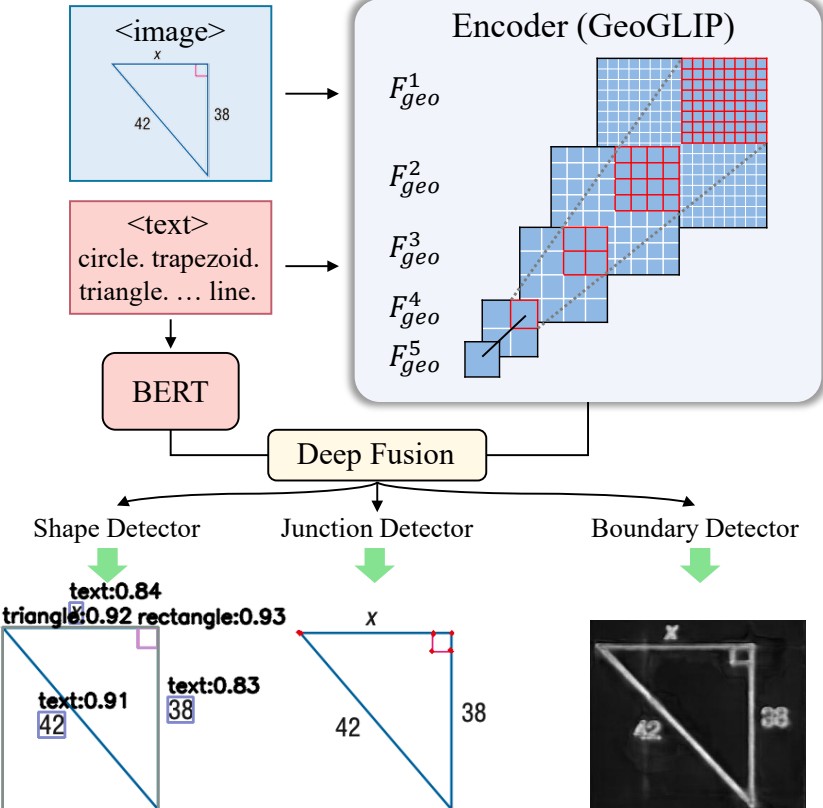

Figure 7: GeoGLIP pipeline. A geometric multi-task detector. GeoGLIP simultaneously detects multiple tasks, including basic geometric shapes, junctions, and boundaries, utilizing multi-scale features to capture fine-grained geometric entities.

with synthetic data could also help reduce the manual effort required. With improved detection performance, we may explore more advanced methods for designing soft prompts, such as object-level prompts. Further refinement of the feature router, such as combining it with interpretable methods to better understand its decision-making process, could also enhance the model's performance. By making the feature router more transparent, we could gain insights into how it selects and prioritizes visual cues, allowing for fine-tuning that aligns better with human intuition and task-specific requirements. This, in turn, would allow for more informed adjustments, leading to better generalization and accuracy in complex mathematical problem-solving scenarios.

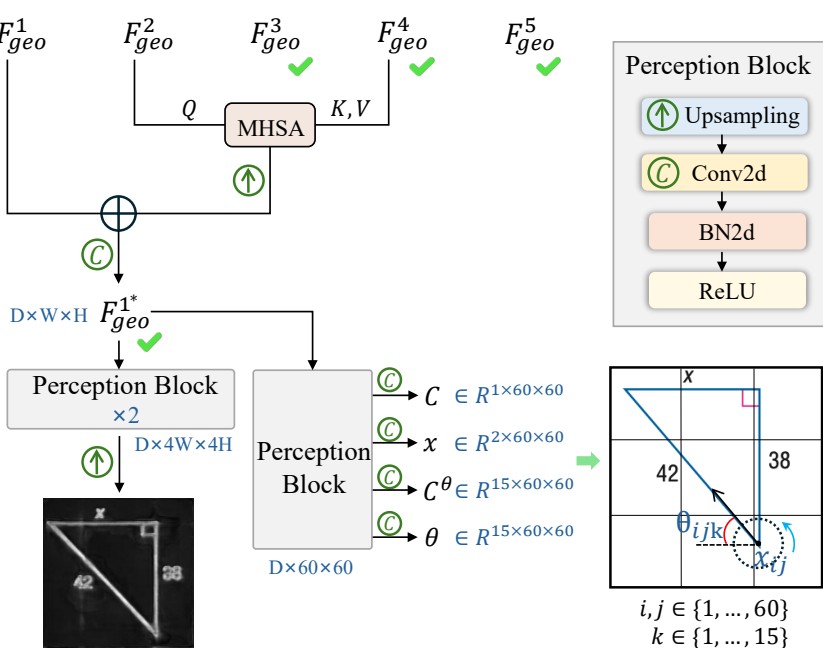

Figure 8: Designs for the junction and boundary detectors: We first use an attention mechanism (MHSA) to fuse two-scale features, followed by upsampling and addition with the highest resolution features, resulting in $F_{geo}^{1*}$. Separate perception blocks are then applied for junction and boundary detection. For junction detection, the detector provides cell confidence ($C$), cell location ($X$), bin confidence ($C^\theta$), and bin orientation ($\theta$). Green check-marked features indicate candidate features for soft prompts, with $D, W, H$ representing channel dim., and spatial resolution (width&height).

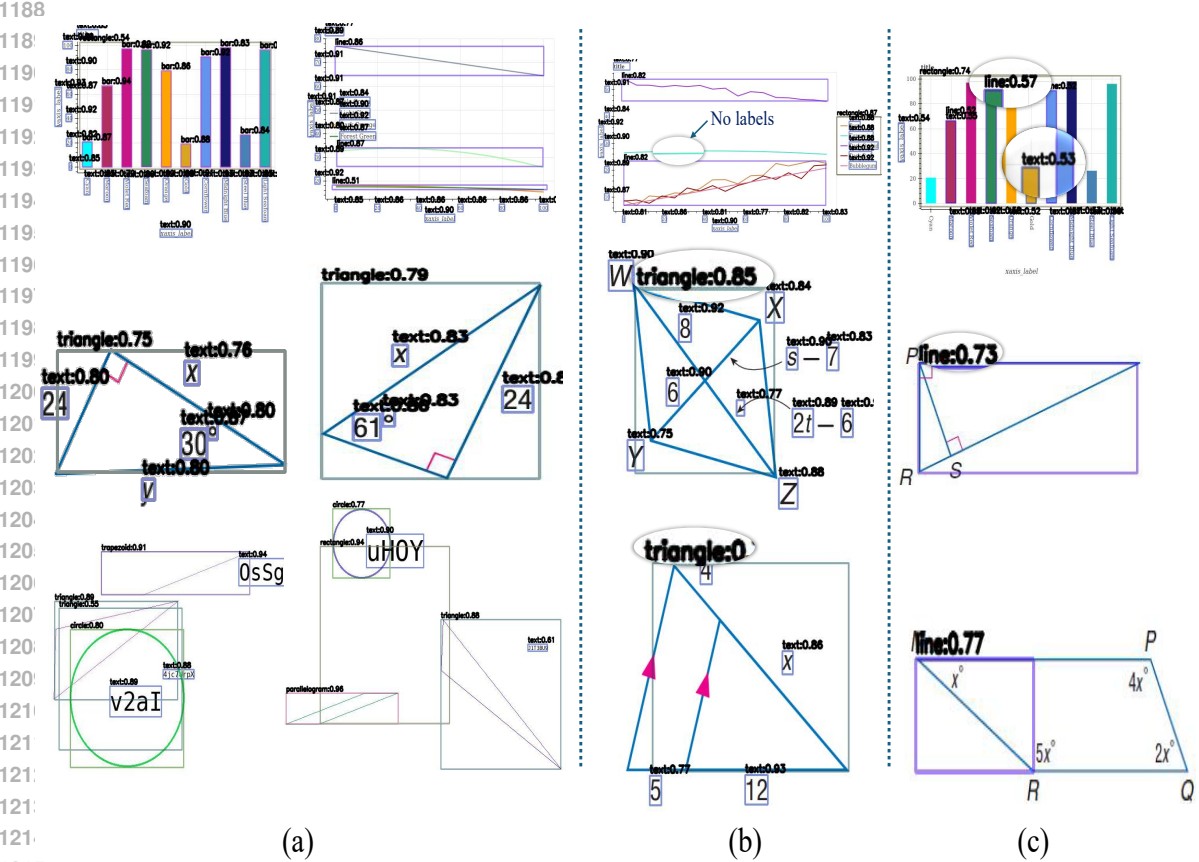

(a)  (b)  (c)

Figure 9: The visualization of shape detection on FigrueQA, Geo170K and our synthetic test dataset. The left panel (a) displays accurate shape detection results generated by GeoGLIP where even small-scale x-ticks are correctly recognized (zoom in 280% for details). GeoGLIP successfully classifies bars in histograms and rectangular shapes in geometric diagrams. The middle panel (b) represents failure cases, with all errors highlighted using a magnifying glass. For instance, in the first row figure, the cyan line is misrecognized, and three crowded lines are incorrectly grouped within a single bounding box. The results in the last panel (c) are generated by the original GLIP, trained on natural images. It is evident that most geometric shapes are misclassified as lines or text, and GLIP struggles to recognize small-scale objects, where GeoGLIP excels.

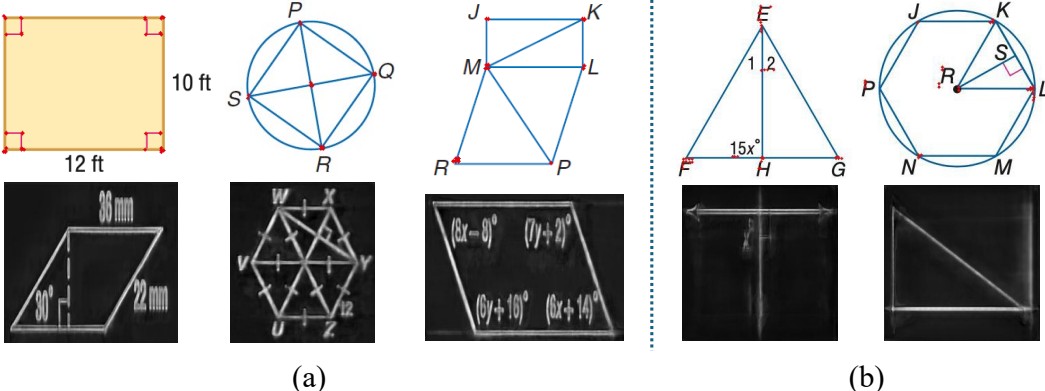

(a)  (b)

Figure 10: The visualization of junction and boundary detection results. The left panel (a) illustrates accurate detections, while the right panel (b) represents failure cases. Junction detection failures frequently exhibit redundant detections.

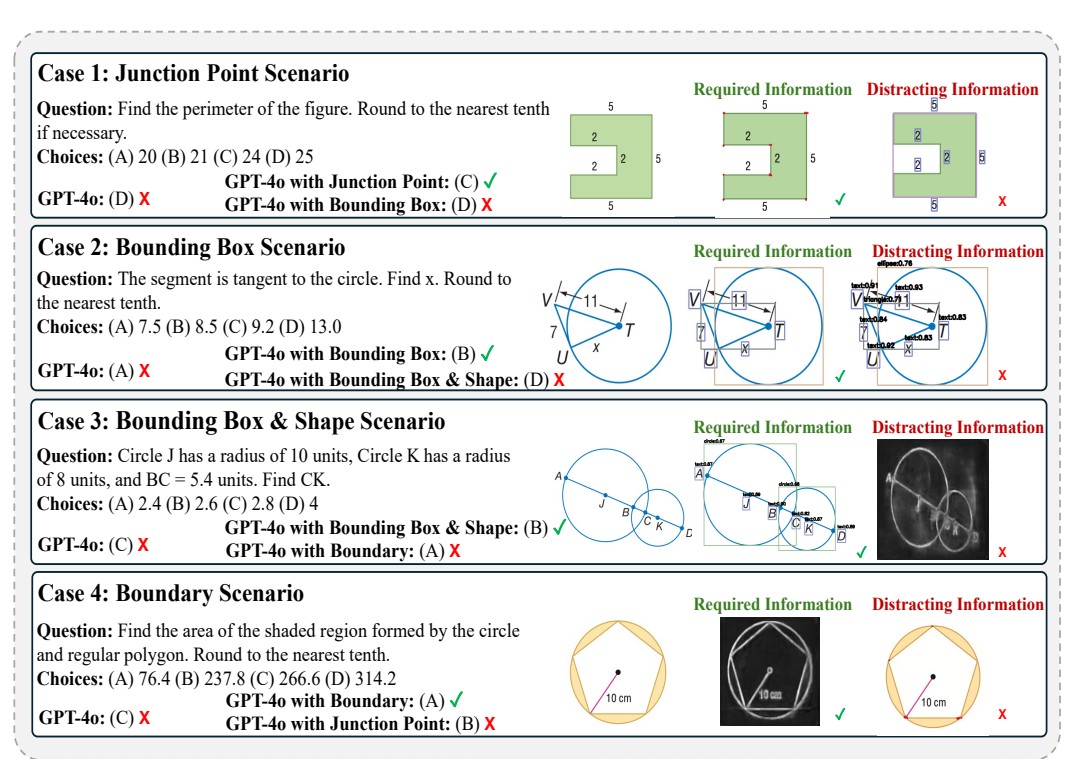

Figure 11: A case study on the Geo170K dataset (Gao et al., 2023a) highlights the importance of providing relevant visual information for each math visual question answer. Zoom in for best view.

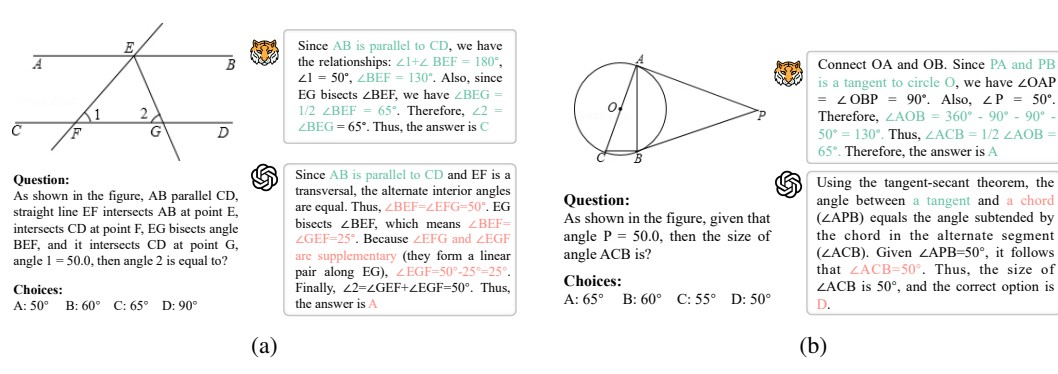

Figure 12: Response comparison of our SVE-Math-Deepseek-7B and GPT-4o. Refer to the main text for detailed analysis. Zoom in for best view.

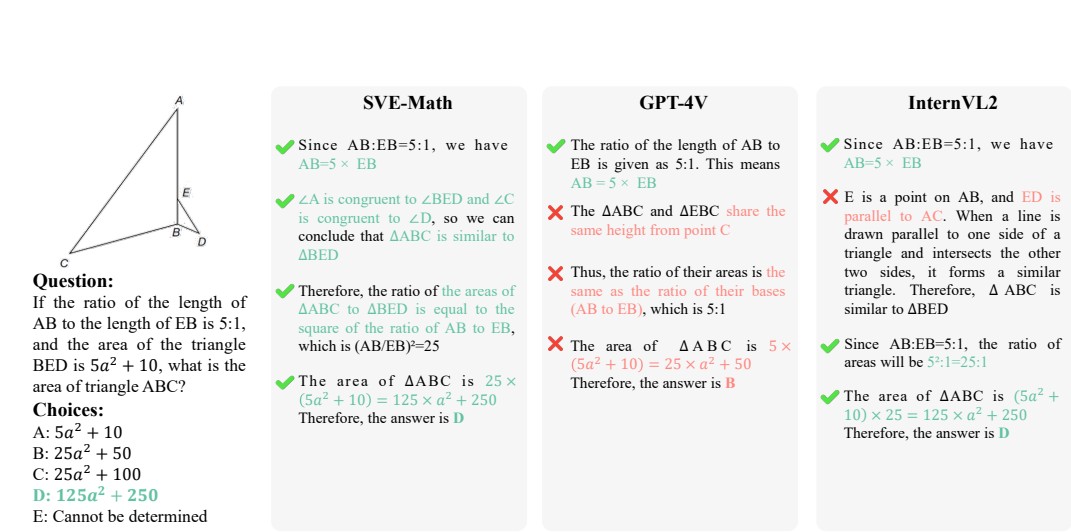

**SVE-Math**

✔ Since AB:EB=5:1, we have AB=5 × EB

✔ ∠A is congruent to ∠BED and ∠C is congruent to ∠D, so we can conclude that ΔABC is similar to ΔBED

✔ Therefore, the ratio of the areas of ΔABC to ΔBED is equal to the square of the ratio of AB to EB, which is (AB/EB)²=25

✔ The area of ΔABC is 25 × (5a² + 10) = 125 × a² + 250 Therefore, the answer is **D**

**GPT-4V**

✔ The ratio of the length of AB to EB is given as 5:1. This means AB = 5 × EB

✘ The ΔABC and ΔEBC share the same height from point C

✘ Thus, the ratio of their areas is the same as the ratio of their bases (AB to EB), which is 5:1

✘ The area of ΔABC is 5 × (5a² + 10) = 25 × a² + 50 Therefore, the answer is **B**

**InternVL2**

✔ Since AB:EB=5:1, we have AB=5 × EB

✘ E is a point on AB, and ED is parallel to AC. When a line is drawn parallel to one side of a triangle and intersects the other two sides, it forms a similar triangle. Therefore, Δ ABC is similar to ΔBED

✔ Since AB:EB=5:1, the ratio of areas will be 5²:1=25:1

✔ The area of ΔABC is (5a² + 10) × 25 = 125 × a² + 250 Therefore, the answer is **D**

**Question:**
If the ratio of the length of AB to the length of EB is 5:1, and the area of the triangle BED is $5a^2 + 10$, what is the area of triangle ABC?
**Choices:**
A: $5a^2 + 10$
B: $25a^2 + 50$
C: $25a^2 + 100$
D: $125a^2 + 250$
E: Cannot be determined

Figure 13: Chain-of-Thought (CoT) response comparison of our SVE-Math-Deepseek-7B, GPT-4V and InternVL2. Refer to the main text for detailed analysis. Zoom in for best view.

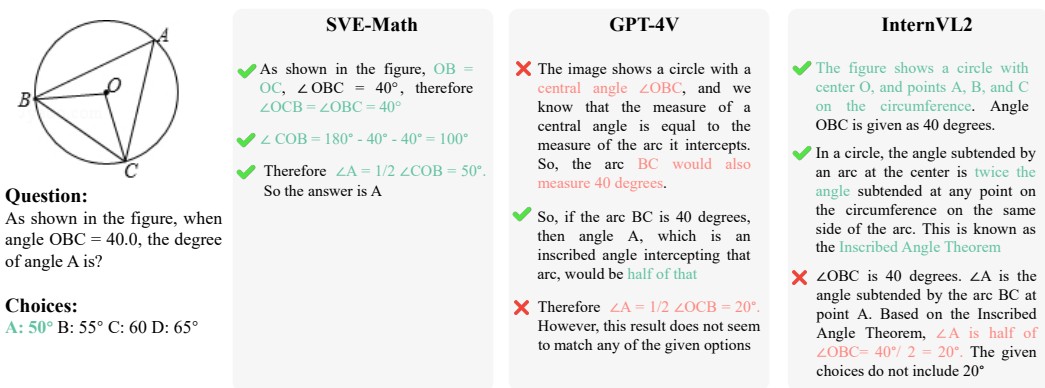

**SVE-Math**

✔ As shown in the figure, OB = OC, ∠ OBC = 40°, therefore ∠OCB = ∠OBC = 40°

✔ ∠ COB = 180° - 40° - 40° = 100°

✔ Therefore ∠A = 1/2 ∠COB = 50°. So the answer is A

**GPT-4V**

✘ The image shows a circle with a central angle ∠OBC, and we know that the measure of a central angle is equal to the measure of the arc it intercepts. So, the arc BC would also measure 40 degrees.

✔ So, if the arc BC is 40 degrees, then angle A, which is an inscribed angle intercepting that arc, would be half of that

✘ Therefore ∠A = 1/2 ∠OCB = 20°. However, this result does not seem to match any of the given options

**InternVL2**

✔ The figure shows a circle with center O, and points A, B, and C on the circumference. Angle OBC is given as 40 degrees.

✔ In a circle, the angle subtended by an arc at the center is twice the angle subtended at any point on the circumference on the same side of the arc. This is known as the Inscribed Angle Theorem

✘ ∠OBC is 40 degrees. ∠A is the angle subtended by the arc BC at point A. Based on the Inscribed Angle Theorem, ∠A is half of ∠OBC= 40°/ 2 = 20°. The given choices do not include 20°

**Question:**
As shown in the figure, when angle OBC = 40.0, the degree of angle A is?

**Choices:**
A: 50° B: 55° C: 60° D: 65°

Figure 14: Chain-of-Thought (CoT) response comparison of our SVE-Math-Deepseek-7B, GPT-4V and InternVL2. Refer to the main text for detailed analysis. Zoom in for best view.