# OpenReview forum: "Open Eyes, Then Reason: Fine-grained Visual Mathematical Understanding in MLLMs"
_ICLR.cc/2025/Conference — ICLR 2025 Conference Withdrawn Submission_

### Official Review · Reviewer_M9gf · 2024-11-02

**Soundness:** 3
**Presentation:** 4
**Contribution:** 2
**Rating:** 5
**Confidence:** 5

**Summary:**

The paper proposes the SVE-Math-7B model to improve the math reasoning skills of current MLLMs. The authors start by analyzing the performance on mainstream models' math reasoning tasks to show the geometric information's effectiveness. Based on the observation, the author proposes the architecture of SVE-Math with a pre-trained GeoGLIP, a fusing connector with dual visual encoders, and further fine-tuning the baseline models. The authors conduct experiments on mainstream math-relative benchmarks such as MathVerse and MathVista and show improvements compared with baselines.

**Strengths:**

1. The paper discovers and discusses the math-solving problem of MLLMs, which is a significant and widely-concerned problem for current MLLMs. The solution with GeoGIP and math-relevant fine-tuning is efficient for the problem.
2. The analysis in Figure 1 clearly shows the drawbacks of LLaVA and GPT-4o, and shows the effectiveness of geometric information.
3. The methods and experiment periods are well organized and easy to follow. The authors conduct experiments on mainstream math datasets and clearly show the results.

**Weaknesses:**

1. The main weakness is that the ablation analysis is not sufficient to demonstrate the improvements of all the components. The author proposes the GeoGLIP, dual visual encoder connector, math-specific finetuning with Geo170K, MathV360K datasets. However, the analysis of such aspects is lacking. The authors only conduct experiments on the design of connectors, which is not the key claim for the contributions, as many papers have used similar fusing approaches for visual encoders. I think the authors could clearly explain where the improvements come from, especially for the GeoGLIP and the math-relevant training datasets.
2. Although the authors show improvements over baselines, the performance for SVE-Math-7B is significantly behind the state-of-the-art models (e.g. more than 60 accuracy on MathVista). I assume the approach proposed by the author is universal, therefore the results of state-of-the-art models are lacking.
3. The effectiveness of GeoGLIP is not confirmed. I wonder how the tiny visual encoder with less than 50M parameters can help the overall learning results. As shown in the visualization results, directly providing geometric-relevant information in a proper manner may also lead to similar performance. The authors could conduct sufficient experiments to explain this issue.

**Questions:**

As stated in the weakness periods, clarifying the issues can better demonstrate the conclusion of the paper.
1. What are the improvements with math-specific datasets?
2. Why using GeoGLIP based on Swin-T is effective for results? As illustrated in the visualization results, the usage of the models provides geometric information, so the authors may provide more comparisons by providing direct geometric results, or directly using GLIP.
3. The results for current models are somehow out-of-date. The authors are encouraged to equip proposed approaches on state-of-the-art level MLLMs.
4. The math problems may already be solved with better data curation or reasoning processes, as many papers have done on such problems. The author could provide explanations and superiority for the proposed methods and provide comparisons with other methods on math problems.
Therefore, based on the weaknesses and questions stated above, I think the paper is below the acceptance threshold in the current situation.

---

> ### Author Response · Authors · 2024-11-28
> **Response to Reviewer M9gf (Part1)**
>
> ##  We thank the reviewer for insightful questions that help refine our work further.
>
> We sincerely thank you for your insightful feedback and constructive suggestions. We greatly appreciate your recognition of the significance of addressing math-solving limitations in MLLMs and the efficiency of our proposed solution, SVE-Math. Below, we provide detailed responses to the specific weaknesses and questions you raised, supported by additional experiments and clarifications.
>
> # 1.  Ablation Analysis and Demonstration of Component Contributions.
> We acknowledge the importance of rigorous ablation studies to isolate the contributions of each component in our model. In our revised manuscript, we provide a detailed analysis that clarifies the individual roles of GeoGLIP, the dual visual encoder connector, and math-specific datasets. The updated results reinforce that GeoGLIP is the primary contributor to the observed performance improvements, aligning with our core motivation. As highlighted in our systematic error analysis (Figure 1), the deficiencies in perceiving geometric primitives significantly impair MLLMs' performance on mathematical reasoning tasks. By addressing these perception gaps, GeoGLIP directly enhances the model's ability through mathematical visual perception content.
>
> **The effectiveness of the proposed GeoGLIP is not validated.**
> We apologize we did not explicitly clarify this point earlier, which has led to the concern regarding whether the improvement observed with our approach primarily stems from the instruction dataset used. To clarify, removing the GeoGLIP encoder—and consequently eliminating the need for the dual visual encoder connector—effectively reduces our SVE-Math-7B to G-LLaVA [Gao et al., 2023a]. Both G-LLaVA and our approach leverage the same LLM backbone (LLaMA2-7B) and the instruction dataset, ensuring that the performance gains are directly attributable to the inclusion of model designs rather than the instruction dataset.  The comparison results are detailed in Tables 1-3. For example, integrating our method into G-LLaVA (our SVE-Math-7B) improves Top-1 accuracy by 7.7\% on MathVerse and 12.3\% on MathVista.
>
>
> **Dual visual encoder connector.**  This ablation is demonstrated by our controlled experiments in Table 5a of Section 4. Specifically, the constant router assigns equal weights to all features, the sparse router  selects a single level of feature map from GeoGLIP, and the soft router assigns learnable dynamic weights. We present the top-1 accuracy results from Table 5a for these configurations. For the sparse router, only the best performance, achieved with the first-level feature map, is shown in the below table.
>
> | Model|Top1 Acc (GeoQA)|
> |:-:|:-:|
> |Constant |62.8|
> |Sparse|64.9|
> |Soft |67.0|
>
> **The comparison of visual encoders.**  We designed a variant that excludes the CLIP visual encoder, relying solely on our soft prompts from the GeoGLIP visual encoder. This resulted in an accuracy drop from 67.0\% to 66.1\%, though it still outperformed the CLIP encoder alone (64.2\%). We leveraged the original hierarchical pyramid features from the GLIP visual encoder (trained on natural image datasets, such as Object365). To ensure a fair comparison, we utilized feature maps with the same resolution: the first layer with the largest resolution and the last three layers with smaller resolutions. This resulted in a performance drop from 67.0\% to 65.3\%, as GLIP lacks sensitivity to geometric details and fails to detect basic geometric shapes, as visualized in Fig. 9.
>
> |Encoder|Model|Top1 Acc (GeoQA)|
> |:-|-:|:-:|
> |Dual encoders|GLIP+CLIP|65.3|
> |Dual encoders|GeoGLIP+CLIP|67.0|
> |single encoder|GeoGLIP|66.1|
> |single encoder|CLIP |64.2|
>
> **Math-specific datasets** (Geo170K and MathV360K) enhance reasoning capabilities, but their effectiveness is significantly amplified when combined with GeoGLIP.
> To further validate GeoGLIP's impact, we conducted experiments comparing its performance against directly incorporating geometric-relevant information (e.g., box/junction coordinates as additional text inputs for instruction fine-tuning) using GLIP. The results fell below the baseline model G-LLaVA, consistent with our observations in Figures b and c. This aligns with the emphasis made in the introduction (lines 107-110), where we noted: "Given the inherent uncertainty in detecting geometric primitives by GeoGLIP, our initial approach utilizes global pyramid feature maps..."
>
> 1. Jiahui Gao et al.,  G-llava: Solving geometric problem with multi-modal large language model, arXiv 2023.

---

> ### Author Response · Authors · 2024-11-28
> **Response to Reviewer  M9gf (Part2)**
>
> # 2.  Performance Relative to State-of-the-Art Models
>
> The performance of SVE-Math, while not surpassing models such as LLaVA-OneVision (72B) with 67.5\% accuracy and InternVL (40B) with 59.9\% accuracy, demonstrates significant resource efficiency and adaptability. Our model achieves strong improvements within a 7B parameter architecture, striking an optimal balance between performance and computational demands. For example, with Qwen2.5-Math-7B, SVE-Math achieves a Top-1 accuracy of 51.3\% on MathVista, surpassing GPT-4V (49.9\%) and showcasing its capacity to compete with larger models. Furthermore, our results indicate that SVE-Math complements reasoning-focused approaches by bridging the gap in visual perception—an area less emphasized in current state-of-the-art designs.
>
> By integrating GeoGLIP with reasoning-optimized LLMs such as DeepSeek-Math-7B, we achieve consistent 5-10\% improvements across benchmarks, reinforcing the generalizability of our approach. These results emphasize that SVE-Math is not merely an alternative but a complementary and modular enhancement that can amplify the capabilities of existing MLLMs.
>
> # 3. Validation of GeoGLIP’s Impact
> GeoGLIP’s lightweight design, based on Swin-T, has been explicitly optimized to enhance visual perception in mathematical tasks. Despite its compact size (less than 50M parameters), GeoGLIP achieves remarkable improvements in performance. The below ablation studies show that removing GeoGLIP  (SVE-Math(-)) results in a significant drop in Top-1 accuracy on MathVista. Its attention-based mechanism enables precise identification and alignment of geometric primitives, junctions, and boundaries, facilitating downstream reasoning tasks.
>
> |Model|Base LLM|All (acc)|
> |:-:|:-:|:-:|
> |G-LLaVA|LLaMA2-7B|25.1|
> |**SVE-Math**|LLaMA2-7B|37.4|
> |SVE-Math(-)|Qwen2.5-7B|44.0|
> |**SVE-Math**|Qwen2.5-7B|51.3|
> |SVE-Math(-)|DeepSeek-7B|42.3|
> |**SVE-Math**|DeepSeek-7B|48.7|
>
> # 4. Applicability to Broader Mathematical Tasks
>
> The modular design of SVE-Math ensures its applicability to a wide range of mathematical tasks beyond geometry-specific problems. Our experimental results demonstrate that the enhanced visual perception capabilities introduced by GeoGLIP significantly benefit tasks that involve non-geometric elements. For instance, experiments on advanced architectures like Qwen2.5-Math-7B and DeepSeek-Math-7B consistently show that GeoGLIP improves overall performance without being restricted to specific types of reasoning challenges.
>
> While our experiments primarily focused on models with 7B parameters, the lightweight and generalizable nature of GeoGLIP ensures its scalability to larger state-of-the-art architectures, including those used in LLaVA-OneVision (72B). Future work will explore further scaling, but the current results already indicate the broad applicability of our approach across diverse mathematical reasoning scenarios.
>
> # 5. Revisions and Enhancements
> To address your concerns and enhance the clarity of our work, we have revised the manuscript to include:
>
> Expanded Ablation Studies: Detailed analysis isolating the contributions of GeoGLIP, the dual encoder connector, and datasets.
> Synthetic Data Descriptions: Comprehensive explanations and visual examples of the synthetic data used for training GeoGLIP.
> Updated Visualizations: Introduction wrapfigure and Appendix Figures 12-14 now include examples demonstrating GeoGLIP’s ability to enhance visual perception and reasoning.
> Benchmark Comparisons: Tables 1-3 compare SVE-Math-Deepseek-7B to state-of-the-art models, emphasizing its resource efficiency and complementary design.

---

> ### Author Response · Authors · 2024-11-30
>
> Esteem Reviewer,
>
> We apologise for our late reply due to  multiple expeirments. Would the reviewer be able to review our rebuttal?  We truly appreciate the reviewer's  feedback and suggestion how to further improve our work.
>
> Bestr regards,
>
>  Authors

---

> ### Comment · Reviewer_M9gf · 2024-11-30
> **Official Comment by Reviewer M9gf**
>
> I appreciate the ablations provided by the authors, especially about GeoGLIP. However, due to the lack of analysis on data curation and relatively low performance, I will keep my negative score as 5.

---

> ### Author Response · Authors · 2024-11-30
> **Response to Reviewer M9gf (Part3)**
>
> We sincerely thank you for taking the time to carefully review our work and for acknowledging the additional ablations we provided regarding GeoGLIP. We understand your concerns about the lack of detailed analysis on data curation and the perceived relatively low performance of our model. We would like to address these points further:
>
> **1. Data Curation Analysis:** regarding data curation in our initial response. To provide a more comprehensive explanation, we have broken this issue into the following two points based on Question 4: **1) Difference between our data curation and other mathematical papers** ("The math problems may already be solved with better data curation or reasoning processes, as many papers have done on such problems"). **2) Superiority of our methods compared to previous mathematical MLLMs** ("The authors could provide explanations and superiority for the proposed methods and provide comparisons with other methods on math problems").
>
>  * Our synthetic visual-centric datasets are fundamentally different from traditional visual-question paired mathematical instruction datasets commonly used in previous mathematical MLLMs. Unlike those methods, which typically rely on GPT-4o to generate diverse prompts and require human intervention to ensure quality, our approach is highly efficient and avoids the labor-intensive and costly process of manual curation. Specifically, our synthetic dataset is programmatically generated using Matplotlib for box-level shape grounding task, and using lightweight, free, publicly available models [Huang et al., 2018;  Verbin et al., 2021] to extract junctions and boundaries as pixel-level ground truth for both our synthetic dataset and public mathematical training images (e.g., Geo170K). This efficient process avoids manual labeling and is explicitly used to train GeoGLIP for shape grounding, junction and boundary detection. Please refer to Section 3.4 and Appendix A.6 for a detailed description of the data curation process. Specifically, Section 3.4 provides an overview of the data curation process, while a flow diagram illustrating the data engine, along with examples of the synthetic diagrams, is presented in Appendix Fig. 6. Data statistics for the synthetic math-specific datasets, including the distribution of geometric shapes and the number of objects per image, are visualized in Figs. 5b and 5c.
>
> > Huang et al., Learning to parse wireframes in images of man-made environments.
>
> > Verbin et al., Field of junctions: Extracting boundary structure at low snr.
>
> *  We have conducted experiments as suggested by the reviewer, directly providing geometric-relevant information to the model. Since no existing mathematical instruction datasets include detailed location information for geometric objects (e.g., bounding box coordinates or junction points), we generated this data by inferring Geo170K training images using GeoGLIP to extract the relevant location information. This information was appended to the special token  <image> in `huam value` supplementary descriptions for each image, using instructions such as: "there is a bounding box at ⟨x, y, w, h⟩ or there is a junction at ⟨x, y⟩ with lines directions  <$\theta$> ".
> When tested on the Geo170K test set of the GeoQA benchmark, the top-1 accuracy dropped from 67.0\% to 63.2\%. This result is close to the variant of our constant router 62.8\% (assigning equal weights to all features, as explained in the dual visual encoder connector in response 1). This performance drop is consistent with our systematic analysis in Figs. b and c: Inaccurate instructions would harm the performance, and relevance is key—excessive visual cues interfere with problem-solving.
>
> We appreciate the reviewer’s suggestion that directly providing geometric-relevant information in a proper manner may also lead to similar performance. Based on our experiments and observations, this proper method would require nearly 100\% accurate grounding results for every mathematical object and highly relevant information tailored to the specific question. However, achieving this would demand significant human resources, including the involvement of mathematical experts.
>
>
> * Our approach instead leverages global pyramid feature maps that encode information ranging from geometry-rich to semantic-rich representations, with their contributions dynamically modulated by the feature router mechanism. Our research underscores the importance of addressing fine-grained visual understanding, a critical bottleneck in visual mathematical reasoning tasks. We hope our work could provide valuable insights for future research and emphasizes the need for more effective integration of fine-grained visual understanding in MLLMs.

---

> > ### Author Response · Authors · 2024-11-30
> > **Response to Reviewer M9gf (Part4)**
> >
> > **1. Data Curation Analysis (Continue):**
> > * Finally, we would like to emphasize the importance of training visual-centric, geometry-aware models with fine-grained box- and pixel-level supervision rather than relying on image-level supervision from contrastive training (e.g., CLIP). That may be the critical reason for the deficient visual perception ability in visual mathematical MLLMs.
> >
> > **2. Relatively Low Performance:**
> >
> > While we agree that our model’s performance still has room for improvement, we would like to highlight that our results represent a significant step forward in addressing the visual perception limitations of multimodal large language models (MLLMs) in mathematical visual reasoning in a small-scale 7B model.
> >
> > MathVerse: SVE-Math-7B achieves 21.2\% accuracy, improving by 7.7\% compared to baseline G-LLaVA-7B, with comparable performance to Math-LLaVA-13B (19.0\%).
> > MathVista: We conducted additional experiments by integrating GeoGLIP with Qwen2.5-Math-7B-Instruct and DeepSeek-Math-7B-Instruct (two of the most advanced mathematical reasoning LLMs currently available).  We evaluate those 7B models on the most challenging MathVista benchmark, achieving 51.3\% and 48.7\% Top-1 accuracy, even surpassing GPT-4V's performance (49.9\%).
> >
> > Our lightweight, geometry-focused design, with less than 50MB\% increase in parameter size and 0.24s\% increase in inference time per image, is orthogonal to approaches emphasizing reasoning, making it a natural complement to such methods. GeoGLIP bridges a critical gap in visual perception for mathematical problems, aligning seamlessly with existing reasoning-optimized models to enhance their capabilities. We will release those model weights, the training, and the inference codes to facilitate the computer vision community.
> >
> > We acknowledge the reviewers' comments regarding the second point of weakness: the state-of-the-art models achieving over 60\% accuracy on MathVista. However, all models achieving such performance either have large-scale parameters (e.g., LLaVA-OneVision with 70B or InternVL-series with 40B/70B parameters) or benefit significantly from large-scale data training, including synthesized knowledge and curated diverse instruction-tuning datasets, such as re-captioned detailed description data, document/OCR data, and multilingual data. While our model uses a smaller-scale dataset for visual-centric training (40K) and 60K + 110K, such as the Geo170K alignment and instruction traning dataset for MLLMs.
> >
> >
> > We fully acknowledge that refining data curation and scaling to larger models (70B) are critical for further enhancing our model. We aim to provide a foundation for addressing these challenges, and your valuable insights have helped us identify the directions where further improvements are most needed.
> >
> > We hope this additional clarification and context address your concerns. We deeply value your feedback and remain committed to improving our work. Should you have any further suggestions or specific points of interest, we are more than willing to address them in the revised version.
> >
> > Thank you again for your thoughtful review and constructive comments.

---

### Official Review · Reviewer_A1wU · 2024-11-02

**Soundness:** 2
**Presentation:** 3
**Contribution:** 2
**Rating:** 5
**Confidence:** 4

**Summary:**

This paper introduces SVE-Math, a Multimodal Large Language Model (MLLM) designed for mathematical question answering. It incorporates a GeoGLIP module to enhance the visual encoder's perception of mathematical elements and utilizes a routing module to prioritize features from CLIP. The training process for SVE-Math consists of three stages: GeoGLIP training, cross-modal alignment, and instruction tuning.

**Strengths:**

1.The approach of enhancing the visual encoder for improved mathematical performance is both innovative and logical.

2.The routing module is well-designed and demonstrates significant performance improvements in the ablation studies. However, I believe that the routing module is not specifically designed for mathematical reasoning tasks and can be applied to a wider range of scenarios.

3.The paper is well-structured and easy to understand.

**Weaknesses:**

1.My main concern is the performance results, which are not particularly impressive. While SVE-Math achieves competitive scores on several benchmarks, the improvements over the previous works are marginal, raising questions about the effectiveness of the approach.

2.Building on the first point, I believe a significant portion of the performance improvement in MLLMs stems from the data used. The scale and quality of training data are critical for MLLMs. Could you elaborate on any unique handling or augmentation techniques applied to the training data?

3.Could the authors provide more explanation of why the routing module is specifically designed for mathematical reasoning tasks? Relying solely on empirical evidence is not sufficient to substantiate this claim.

**Questions:**

Please refer to weakness. Can the proposed methods be applied to other mathematical problems beyond geometric figures and problems?

---

> ### Author Response · Authors · 2024-11-28
> **Response to Reviewer A1wU (Part1)**
>
> ##We thank the reviewer for insightful questions that help refine our work further.
> # 1. Performance Results and Marginal Improvement
> We appreciate the reviewer's comment, but we respectfully disagree with the perception that the performance improvements are marginal. Under identical configurations, including the same base LLM (LLaMA2-7B) and model size (7 billion parameters), our model demonstrates significant performance improvements:
>
> MathVerse: SVE-Math-7B achieves 21.2\% accuracy, improving by 7.7\% compared to G-LLaVA-7B, with comparable performance to Math-LLaVA-13B (19.0\%).
> MathVista: We conducted additional experiments by integrating GeoGLIP with Qwen2.5-Math-7B-Instruct and DeepSeek-Math-7B-Instruct (two of the most advanced mathematical reasoning LLMs currently available).  We evaluate those 7B models on the most challenging MathVista benchmark, achieving 51.3\% and 48.7\% Top-1 accuracy, even surpassing GPT-4V's performance (49.9\%).
>
> These results are achieved under controlled conditions, ensuring that performance gains arise from the inclusion of GeoGLIP rather than data scale or quality differences. Our lightweight, geometry-focused design, with less than 50MB\% increase in parameter size and 0.24s\% increase in inference time per image, is orthogonal to approaches emphasizing reasoning, making it a natural complement to such methods. GeoGLIP bridges a critical gap in visual perception for mathematical problems, aligning seamlessly with existing reasoning-optimized models to enhance their capabilities. We will release those model weights, the training, and the inference codes to facilitate the computer vision community.
>
> # 2. Data Contributions and Generalization
> The synthetic data used for training GeoGLIP is designed to efficiently improve geometric perception without introducing dataset biases. Unlike manually curated instruction datasets for training MLLMs, our synthetic dataset is programmatically generated using Matplotlib for box-level shape grounding task, and using off-the-shelf models to extract junctions and boundaries as pixel-level ground truth for both our synthetic dataset and public mathematical training images (e.g., Geo170K). This efficient process avoids manual labeling and is explicitly used to train GeoGLIP for shape grounding, junction and boundary detection.
>
> Our ablation studies confirm that improvements in SVE-Math do not stem solely from the training data. When G-LLaVA and SVE-Math-7B are trained on the same datasets, integrating GeoGLIP to G-LLaVA (our SVE-Math-7B) leads to consistent performance gains: 1) MathVerse: 7.7\% improvement over G-LLaVA; 2) MathVista: 12.3\% improvement.
> Furthermore, the modular design of GeoGLIP enables generalization across diverse LLM backbones. Experiments with Qwen2.5-Math-7B and DeepSeek-Math-7B demonstrate 6-7\% improvements across benchmarks, highlighting GeoGLIP’s adaptability to advanced architectures (SVE-Math vs. SVE-Math(-)).
>
> |Model|Base LLM|All (acc)|
> |:-:|:-:|:-:|
> |G-LLaVA|LLaMA2-7B|25.1|
> |**SVE-Math**|LLaMA2-7B|37.4|
> |SVE-Math(-)|Qwen2.5-7B|44.0|
> |**SVE-Math**|Qwen2.5-7B|51.3|
> |SVE-Math(-)|DeepSeek-7B|42.3|
> |**SVE-Math**|DeepSeek-7B|48.7|
>
> # 3. Design and Applicability of the Feature Router
> The routing module dynamically prioritizes geometry-rich features from GeoGLIP and semantic-rich features from CLIP, selectively enhancing visual perception. While the module is broadly applicable, it is specifically designed to address challenges in mathematical reasoning in our paper:
>
> Selective Filtering: Mathematical tasks often involve irrelevant visual elements that hinder reasoning (as shown in Fig. 1 of the paper). The routing module ensures only relevant cues are passed to the reasoning components, addressing this bottleneck.
> Empirical Validation: Ablation studies (Table 5) show a 4-6\% accuracy improvement attributable to the routing mechanism, confirming its effectiveness.
> We acknowledge the reviewer’s suggestion for further theoretical substantiation of the routing module’s design. This is a valuable direction for future work, where we aim to develop formal frameworks for task-specific feature prioritization.
>
> # 4. Applicability Beyond Geometric Problems
> GeoGLIP is not limited to geometric problems/figures . Its lightweight, modular design enhances visual perception in diverse mathematical tasks, as evidenced by its consistent performance gains across multiple benchmarks, particularly in MathVista, which spans a diverse array of mathematical tasks, including Textbook Question Answering (TAQ), Visual Question Answering (VQA), Figure Question Answering (GQA), and icon-based visual question answering (IconQA). Integration with reasoning-optimized LLMs (e.g., DeepSeek-Math-7B) demonstrates its general applicability, yielding improvements in both visual and non-visual tasks (math word problem, MWP).

---

> ### Author Response · Authors · 2024-11-28
> **Response to Reviewer A1wU (Part2)**
>
> # 4. Applicability Beyond Geometric Problems (Continue)
>
> Additionally, SVE-Math supports Chain-of-Thought (CoT) reasoning by combining improved visual perception with logical inference. Examples provided in the revised manuscript (Introduction wrapfigure and Appendix Figures 12-14) demonstrate how SVE-Math effectively recognizes mathematical elements and leverages CoT reasoning to address problems that combine visual and textual inputs.
>
> # 5. Revisions and Clarifications
> To address the reviewer’s concerns and enhance the clarity of our paper, the revised manuscript includes:
>
> **Expanded Data Descriptions:** Detailed explanations of synthetic data generation and examples of annotated diagrams in Section 3.4 and Appendix A.6. A flow diagram illustrating the data engine, along with examples of the synthetic diagrams, is presented in Appendix Fig. 6. Additionally, data statistics for the synthetic math-specific datasets, including the distribution of geometric shapes and the number of objects per image, are visualized in Fig. 5b and Fig. 5c.
> **Routing Module Insights:** A dedicated subsection discussing the module’s design, functionality, and empirical contributions.
> Qualitative Examples: Visualizations of model outputs (Introduction wrapfigure and Appendix Figures 12-14) showcasing SVE-Math’s ability to integrate visual perception with reasoning.

---

> ### Author Response · Authors · 2024-12-02
>
> We would like to sincerely thank you for your valuable feedback and thoughtful suggestions on our paper. We have carefully considered your recommendations, addressed your concerns, and updated both the revised paper and our responses accordingly.
>
> If you have any further questions or concerns, we would be delighted to address them promptly. Your insights are crucial to us, and we deeply appreciate the time and effort you have dedicated to reviewing our work.
>
> To summarize our main contributions:
>
> We systematically identify and analyze the impact of visual recognition errors on the mathematical reasoning performance of MLLMs, highlighting the critical role of accurately perceiving geometric primitives. This new aspect is orthogonal to existing methods focused on improving reasoning.
>
> We designed GeoGLIP, a lightweight, geometry-aware visual model with multitask learning capabilities, including shape grounding, junction detection, and boundary detection. GeoGLIP integrates seamlessly with diverse LLM backbones without requiring modifications to their reasoning components. Despite adding less than a 50MB increase in parameter size and only a 0.24s increase in inference time per image, and without relying on additional mathematical instruction datasets, our approach achieves an 8–12\% improvement in top-1 accuracy compared to the baseline (using LLaMA2-7B as the base LLM).
>
> When paired with advanced LLMs like DeepSeek and Qwen, our 7B model achieves performance comparable to GPT-4V, with 51.3\% and 48.7\% on the challenging MathVista benchmark, versus 49.9\% for GPT-4V.  While our 7B model does not surpass state-of-the-art MLLMs with 40B/70B parameters achieving over 60\% accuracy, integrating GeoGLIP into such large-scale LLMs is currently computationally prohibitive due to our limited resources.   We hope this work inspires further research into more effective fine-grained visual understanding in MLLMs. To support the community and assist other researchers in scaling our method to larger models and datasets,
> we will release the model weights, training scripts, and inference codes to facilitate broader adoption and experimentation.

---

### Official Review · Reviewer_yC4U · 2024-11-03

**Soundness:** 3
**Presentation:** 3
**Contribution:** 2
**Rating:** 5
**Confidence:** 4

**Summary:**

To address the limitations of multimodal large language models (MLLMs) in solving math problems involving images, this paper proposes a Selective Vision-Enhanced Mathematical MLLM. It leverages a geometric-grounded vision encoder and a feature router to help MLLMs better comprehend mathematical image features, thereby improving their performance on math problems with visual components.

**Strengths:**

1. The paper clearly articulates the problem it aims to address, and the overall writing is easy to follow.

2. The paper enhances MLLM's ability to recognize mathematical images and solve math problems by introducing geometry-rich visual information, achieving improvements on several benchmarks.

**Weaknesses:**

1. Using more detailed visual features for solving math problems is an intuitive idea, as is combining geometric and semantic features at different levels. However, you should conduct additional ablation studies to validate the effectiveness of this approach. For instance, consider using vision encoders from other similar models on your dataset/training your model on the training data of other models.

2. In Table 1, some experimental results differ from those provided in the official MathVerse table. For example, you show the cot-e score for SPHINX-Plus and the w/o score for SPHINX-MOE. When comparing with other models on the same benchmark, you should ensure thorough variable control.

3. You mention using synthetic data, but the paper does not include any description, details, or examples of the synthetic data generation process.

4. The paper does not present any output examples from the model.

5. As a “data collection-model training-benchmark testing” type of paper, the performance improvements on benchmarks are minimal in the absence of novelty.

**Questions:**

1. In terms of writing, the paper’s section distribution could be improved. You should allocate some space to introduce synthetic data, dedicate more space to ablation studies to validate the method's effectiveness, and reduce the length of the Methods section.

2. Please provide more details and examples of the synthetic data.

3. Please provide examples of the model’s outputs to demonstrate its ability to recognize geometric elements and Chain-of-Thought (CoT), as you compared cot-e performance with some models in Table 1.

4. In the Introduction, you mentioned a finding: instructing MLLMs with fine-grained visual information improves top-1 accuracy compared to providing only worded questions, while providing all visual cues for solving a math question decreases accuracy. How does your approach—primarily by introducing more geometry-rich visual information—address the issue highlighted by this finding?

---

> ### Author Response · Authors · 2024-11-28
> **Response to Reviewer yC4U (Part1)**
>
> ## We sincerely thank the reviewer for their thoughtful feedback, which provides valuable insights for refining our work. We address your concerns and questions below, supplemented by additional experiments and clarifications in our revised submission.
>
> # 1. Data collection-model training-benchmark testing.
>  We would like to clarify the distinction between our synthetic math-specific datasets and traditional mathematical instruction datasets. We do not create or use any additional self-generated instruction datasets beyond the publicly available Geo170K and MathV360K datasets for MLLM training. Instead, our synthetic samples, annotated with box/pixel-level details, are exclusively utilized to train the GeoGLIP visual encoder.  Compared to constructing mathematical instruction datasets, our synthetic data generation process is significantly more efficient and resource-friendly. It does not require manual labeling, as all data can be programmatically generated, e.g., through the Matplotlib Python library. In contrast, constructing instruction datasets often relies on GPT-4o to create diverse prompts and necessitates human intervention, making the process labor-intensive and costly. Moreover, training the lightweight, visual-centric GeoGLIP involves straightforward training recipes. In comparison, instruction tuning for MLLMs requires intricate configurations, such as carefully curated batch sizes and learning rates, as noted in [Shengbang et al., 2024].
> > Shengbang Tong et al., Cambrian-1: A Fully Open, Vision-Centric Exploration of Multimodal LLMs, arXiv 2024.
>
> # 2. Novelty of Using Geometry-Rich Features.
> While leveraging geometry-rich visual information might appear intuitive, we argue that our method goes beyond combining geometric and semantic features. Our key contribution lies in the introduction of GeoGLIP, a domain-specific geometric-aware visual encoder with hierarchical feature pyramids dynamically weighted by a feature router mechanism. GeoGLIP is equipped with multi-task learning capabilities, including shape grounding, junction detection, and boundary detection (Section 3.2). These innovations directly address the bottleneck of fine-grained visual perception in mathematical reasoning, as demonstrated by our systematic error analysis (Fig. 1). Notably, existing models like GPT-4V misinterpret geometric primitives in 70\% of cases, as highlighted in our study. In response to reviewer suggestions, we compare the outputs of our model with other advanced MLLMs (GPT-4o, GPT-4V and InterVL2). For instance, as shown in the Sec. Introduction, GPT-4o struggles to accurately perceive mathematical elements, impairing its ability to narrate their relationships during the reasoning process. By integrating GeoGLIP, our SVE-Math effectively grounds geometric elements and their positional relationships, enabling accurate reasoning.
> # 3. Additional Ablation Studies.
> We appreciate the reviewer's insightful question. We have conducted comprehensive ablation experiments. These include:
>
> **1) Consider using vision encoders from other similar models on our dataset.**
> We would like to clarify why we chose the GLIP detector: GLIP is an open-set object detector capable of identifying arbitrary classes by matching visual features with corresponding language embeddings. Unlike traditional object detectors with learnable classification weights, GLIP's multi-modal architecture offers greater generality to novel objects and surpasses previous traditional object detectors. In response to the reviewer's concern,, we replaced GLIP with another open-set object detector, Grounding DINO [Liu et al., 2024], and fine-tuned it on our math-specific dataset. We visualized the detection results, as we did for GeoGLIP in Fig. 9 and Fig. 10, which show that Grounding DINO fails to effectively detect small-scale geometric primitives. Upon debugging the code and training setup, we hypothesize this limitation is due to architectural differences. Grounding DINO, as a DETR-based detector, relies solely on the last-layer features of its visual encoder for cross-attention with query embeddings for fianl detection. In contrast, GLIP, as a Faster-RCNN-based detector, utilizes multi-scale features for both bounding box regression and classification, offering better small-object detection capabilities. When integrating the fine-tuned Grounding DINO encoder into our pipeline, the top-1 accuracy on the GeoQA benchmark dropped from 67.0\% to 66.1\%, further supporting GLIP's advantages for our tasks.

---

> ### Author Response · Authors · 2024-11-28
> **Response to Reviewer yC4U (Part2)**
>
> # 3. Additional Ablation Studies (Continue).
> **2) Training our model on the training data of other models.** We are the first to construct a math-specific dataset, including geometric bounding box annotations, as well as junction and boundary annotations. Thus, we leveraged the original hierarchical pyramid features from the GLIP visual encoder (trained on natural image datasets, such as Object365 and MSCOCO). To ensure a fair comparison, we utilized feature maps with the same resolution: the first layer with the largest resolution and the last three layers with smaller resolutions. This resulted in a performance drop from 67.0\% to 65.3\%, as GLIP lacks sensitivity to geometric details and fails to detect basic geometric shapes, as visualized in Fig. 9.
>
>
> > Liu et al.,  Grounding dino: Marrying dino with grounded pre-training for open-set object detection, ECCV 2024.
>
> > DETR: Carion et al., End-to-end object detection with transformers, ECCV 2020.
>
> > Faster R-CNN: Ren et al., Faster R-CNN: Towards real-time object detection with region proposal networks, NeurIPS 2015.
>
> # 3. Comparison and Control in Table 1
> We appreciate your observation regarding discrepancies in Table 1. We have carefully revisited the MathVerse dataset and revalidated all results under consistent experimental setups, ensuring strict variable control. In Table 1, our model reports direct accuracy under the 'w/o' scores, instead of using the CoT evaluation strategy. Additionally, we have updated the corrected accuracy for other models.
>
> # 4. Synthetic Data Generation
> Thank you for highlighting the need for elaboration on synthetic data. We now provide a detailed explanation in Section 3.4 and Appendix A.6.  Our synthetic dataset is programmatically generated using Matplotlib for box-level shape grounding task, and using off-the-shelf models to extract junctions and boundaries as pixel-level ground truth for both our synthetic dataset and public mathematical training images (e.g., Geo170K). This efficient process avoids manual labeling and is explicitly used to train GeoGLIP for shape grounding, junction and boundary detection. A flow diagram illustrating the data engine, along with examples of the synthetic diagrams, is presented in Appendix Fig. 6. Additionally, data statistics for the synthetic math-specific datasets, including the distribution of geometric shapes and the number of objects per image, are visualized in Fig. 5b and Fig. 5c.
>
> # 5. Examples of Model Outputs
> We have added qualitative examples of SVE-Math’s outputs to the revised paper (Figures 12-14). These examples illustrate its ability to: 1) Accurately recognize geometric primitives and positional relationships, facilitating clear and logical mathematical reasoning in the model's responses, and 2) Apply Chain-of-Thought (CoT) reasoning to effectively integrate visual and textual information.
>
> Refer to Section A.5 for more analysis.
>
> # 6. Addressing Accuracy Drop with Excess Visual Cues
> Our approach directly addresses the paradox of excess visual information lowering accuracy, as noted in GPT-4V’s performance (Fig. 1c). By dynamically adjusting the contributions of visual features through the feature router, our method filters irrelevant cues, providing only contextually relevant visual prompts. This selective enhancement improves reasoning without introducing noise, as demonstrated by our controlled experiments in Table 5a of Section 4. Specificlaly, the constant router assigns equal weights to all features, the sparse router  selects a single level of feature map from GeoGLIP, and the soft router assigns learnable dynamic weights. We present the top-1 accuracy results from Table 5a for these configurations. For the sparse router, only the best performance, achieved with the first-level feature map, is shown in the below table.
> |Model|Top1 Acc (GeoQA)|
> |:-:|:-:|
> |Constant |62.8|
> |Sparse|64.9|
> |Soft |67.0|
>
> # 7. Minimal Performance Gains
> While the reviewer perceives performance gains as minimal, we respectfully disagree. Under identical configurations, including the same base LLM (LLaMA2-7B) and model size (7 billion parameters), our model demonstrates significant performance improvements.  For example, as detailed in Tables 1 and 2, integrating our method into G-LLaVA (our SVE-Math-7B) improves Top-1 accuracy by 7.7\% on MathVerse and 12.3\% on MathVista. Other mathematical MLLMs often rely on larger-scale models or more advanced LLMs with stronger reasoning capabilities. Comparing our 7B model, based on the standard LLaMA2-7B, to these MLLMs may not provide a fully equitable evaluation. We conducted additional experiments by integrating GeoGLIP with Qwen2.5-Math-7B-Instruct and DeepSeek-Math-7B-Instruct (two of the most advanced mathematical reasoning LLMs currently available).

---

> > ### Author Response · Authors · 2024-11-28
> > **Response to Reviewer yC4U (Part3)**
> >
> > # 7. Minimal Performance Gains (Continue)
> >
> > We evaluate those 7B models on the most challenging MathVista benchmark, achieving 51.3\% and 48.7\% Top-1 accuracy, even surpassing GPT-4V's performance (49.9\%).  Again, we observe a consistent 6\%-7\% improvement compared to the variant excluding GeoGLIP featurs as additional visual promtps (SVE-Math(-)).  These results reaffirm the complementary nature of the GeoGLIP visual encoder with reasoning abilities and highlight its generalizability benefits across diverse architectures. We will release those model weights, the training, and the inference codes to facilitate the computer vision community.
> >
> > |Model|Base LLM|All (acc)|
> > |:-:|:-:|:-:|
> > |G-LLaVA|LLaMA2-7B|25.1|
> > |**SVE-Math**|LLaMA2-7B|37.4|
> > |SVE-Math(-)|Qwen2.5-7B|44.0|
> > |**SVE-Math**|Qwen2.5-7B|51.3|
> > |SVE-Math(-)|DeepSeek-7B|42.3|
> > |**SVE-Math**|DeepSeek-7B|48.7|
> >
> > # 8. Section Organization
> > We appreciate the suggestion to improve section distribution. In the revised manuscript:
> >
> > The Methods section has been streamlined, with detailed training protocols moved to the appendix.
> > Synthetic data descriptions and model output examples have been expanded in the main text.

---

> > > ### Comment · Reviewer_yC4U · 2024-11-29
> > >
> > > The authors have indeed addressed some of the issues highlighted in the review by improving the clarity and coherence of their paper. The revised version is more fluent and makes it easier to grasp the key points.
> > >
> > > Additionally, the authors have conducted the missing experiments mentioned in the review.
> > >
> > > As a result, I am raising my score to 5.

---

> ### Author Response · Authors · 2024-11-30
>
> We are very glad that our response has resolved your concerns. We thank the reviewer for valuable comments helping us improve our work. We will address them all in the revised manuscript.
>
> ~~Is there anything else we could improve or refine in order to obtain score 6?~~
>
> Please let us know if there are any additional technical aspects you believe we could refine further.
>
> Best regards,
>
> Authors

---

### Official Review · Reviewer_FTUd · 2024-11-04

**Soundness:** 2
**Presentation:** 3
**Contribution:** 2
**Rating:** 5
**Confidence:** 5

**Summary:**

The paper first identifies visual recognition errors prevalent of current MLLMs by a pilot study. Then the paper introduces GeoGLIP, a vision encoder specifically trained to identify geometric elements in the image. The feature from the trained geometric vision encoder is later merged with the feature of the original CLIP vision encoder, aiming at more precise geometry perception. The authors prove the effectiveness of their method by evaluating on various benchmarks.

**Strengths:**

+ The paper proposes a novel perspective that the errors of visual mathematical problems come from poor visual perception.
+ The three training tasks of GeoGLIP closely matches the analysis in Fig. 1. The paper is self-contained and well-written.

**Weaknesses:**

+ The major concern is whether addressing the visual perception error is sufficient for the mllm to correctly solve these tasks. The visual mathematical questions also require advanced reasoning capability, especially merging both the visual and textual information. Only correctly identifying the graph seems to be far enough to solve a mathematical problem. Detecting the texts, shapes or curves in the graph does not necessarily suggest the model understands the element. How much GeoGLIP actually helps in understanding and reasoning seems marginal. The pilot study shown in Fig. 1 also only analyze the error of visual descriptions, while neglecting other potential core problems of MLLM for visual mathematical questions.
+ The effectiveness of the proposed GeoGLIP is not validated. The authors need to report the performance of the model trained with same instruction data only without the GeoGLIP encoder to illustrate the improvement brought by it. Otherwise, the improvement may be from the Geo170K data.
+ The overall performance advantages of SVE-Math compared to previous works are not very obvious.

**Questions:**

+ How much more computational cost and inference time is introduced by GeoGLIP?

---

> ### Author Response · Authors · 2024-11-28
> **Response to Reviewer FTUd (Part1)**
>
> ## We thank the reviewer for helpful comments.
> # 1. Response to Reviewer Concern: Addressing the visual perception error is insufficient for the MLM to correctly solve these tasks.
> Thank you for raising this important concern about the insufficiency of addressing visual perception errors in MLLMs for solving mathematical tasks, especially those requiring the integration of visual and textual information and advanced reasoning. We appreciate the opportunity to clarify our contributions and discuss the interplay between these capabilities.
>
> **Key Clarifications on Visual and Reasoning Abilities.**
> We conceptualize the capabilities required for visual mathematical reasoning into three core abilities: visual perception, visual understanding, and text-world reasoning.  Visual perception refers to the ability to recognize basic geometric primitives (shapes, bounding box locations, and boundaries), which serve as the building blocks of mathematical diagrams; Visual understanding involves aligning visual features with their corresponding textual embeddings—addressing the reviewers' concern about how the model comprehends geometric elements; text-world reasoning refers to the model's capacity to follow logical reasoning steps for providing the final answer.
>
> While prior research has predominantly focused on the last reasoning abilities by constructing large-scale mathematical visual instruction datasets and fine-tuning MLLMs on mathematical domains, our work takes an orthogonal approach by emphasizing visual perception as a critical yet underexplored foundation for effective mathematical solving.
>
> **Addressing Visual Perception and Visual Understanding Gaps.** Our study is the first to systematically analyze the impact of fine-grained visual cues on MLLM performance for mathematical tasks. Figure 1 highlights that visual recognition errors are pervasive in MLLMs and significantly degrade their mathematical reasoning capabilities. These errors stem from deficiencies in both visual perception and visual understanding.
>
> To address these challenges, our contributions include:
>
> 1)  A geometric visual encoder (Geometric-Grounded Language-Image Pre-training duded as GeoGLIP): This encoder enhances perception by accurately identifying basic geometric shapes, junctions, and boundaries, thereby addressing the foundational layer of visual recognition.
>
> 2) Initial methods for visual understanding: In Section 3.3, we describe a straightforward connector design leveraging the simple and effective MLP projectors (linear layer + GELU + linear layer), similar to LLaVA. This approach is a starting point for addressing visual-textual alignment.
>
> **Future Directions for Enhanced Visual Understanding.** We acknowledge that more sophisticated strategies could further improve visual understanding, especially for directly aligning individual geometric objects with their corresponding textual descriptions. Achieving this would require a visual tokenizer capable of representing each object as individual visual tokens, rather than relying on the simple grid-based partitioning used in current visual encoders, which fails to guarantee the integrity of whole objects. To the best of our knowledge, such a visual tokenizer does not currently exist, making its development another promising direction for future research.
>
> We thank the reviewer for pointing out these critical aspects and hope this response clarifies our contributions and the scope of our research.
>
> # 2. How much GeoGLIP actually helps in understanding and reasoning seems marginal.
> Thank you for pointing this out. As noted by the first response, the main goal of GeoGLIP is to enhance the visual perception capabilities of MLLMs in a way that complements the following understanding and reasoning abilities.  To support our motivation, we conducted an additional systematic analysis to quantify the impact of visual perception ability on mathematical reasoning tasks. By manually correcting each visual perception error identified in Figure 1, we observed an overall approximate 12\% increase in accuracy on corresponding mathematical questions. A detailed bar plot of these statistics is included in Figure 5a of the revised paper, providing direct evidence of the importance of enhanced visual perception ability. We also provide the model response outputs in Sections Introduction and A.5 of revisions. Further evidence for the benefits of improved perception is shown in Tables 1–3. Compared to the baseline model G-LLaVA, which shares the same reasoning process and LLM backbone (LLaMA2-7B), the only change in our approach is the integration of the GeoGLIP features. The improvement is significant. For example, as detailed in Tables 1 and 2, integrating our method into G-LLaVA (our SVE-Math-7B) improves Top-1 accuracy by 7.7\% on MathVerse and 12.3\% on MathVista, underscoring the substantial contribution of enhanced visual perception to overall performance.

---

> ### Author Response · Authors · 2024-11-28
> **Response to Reviewer FTUd (Part2)**
>
> # 3. The effectiveness of the proposed GeoGLIP is not validated.
> We apologize we did not explicitly clarify this point earlier, which has led to the concern regarding whether the improvement observed with our approach primarily stems from the instruction dataset used (Geo170K). To clarify, removing the GeoGLIP encoder degenerates our SVE-Math-7B to G-LLaVA [Gao et al., 2023a]. Both G-LLaVA and our approach leverage the same LLM backbone (LLaMA2-7B) and the Geo170K instruction dataset, ensuring that the performance gains are directly attributable to the inclusion of the GeoGLIP encoder rather than the instruction dataset.  The comparison results are detailed in Tables 1-3. Notably, our SVE-Math-7B even achieves comparable performance to Math-LLaVA-13B on MathVerse (19.0\% vs. 21.2\%),  particularly excelling in the 'visual-only' scenario (16.4\% vs. 20.3\%). This scenario strips away the entire textual input, conveying the problem solely through the diagram.
>
> > Jiahui Gao et al.,  G-llava: Solving geometric problem with multi-modal large language model, arXiv 2023.
>
> # 4. The overall performance advantages of SVE-Math compared to previous works are not very obvious.
>
> Thank you for the feedback. We politly disagree. As highlighted in the above responses, under identical configurations, including the same base LLM (LLaMA2-7B) and model size (7 billion parameters), our model demonstrates significant performance improvements. Other mathematical MLLMs often rely on larger-scale models or more advanced LLMs with stronger reasoning capabilities. Comparing our 7B model, based on the standard LLaMA2-7B, to these MLLMs may not provide a fully equitable evaluation. In response to the reviewer's concern, we conducted additional experiments by integrating GeoGLIP with Qwen2.5-Math-7B-Instruct and DeepSeek-Math-7B-Instruct (two of the most advanced mathematical reasoning LLMs currently available).  We evaluate those 7B models on the most challenging MathVista benchmark, achieving 51.3\% and 48.7\% Top-1 accuracy, even surpassing GPT-4V's performance (49.9\%).  Again, we observe a consistent 6\%-7\% improvement compared to the variant excluding GeoGLIP featurs as additional visual promtps (SVE-Math(-)).  These results reaffirm the complementary nature of the GeoGLIP visual encoder with reasoning abilities and highlight its generalizability benefits across diverse architectures. We will release those model weights, the training, and the inference codes to facilitate the computer vision community.
>
> |Model|Base LLM|All (acc)|
> |:-:|:-:|:-:|
> |G-LLaVA|LLaMA2-7B|25.1|
> |**SVE-Math**|LLaMA2-7B|37.4|
> |SVE-Math(-)|Qwen2.5-7B|44.0|
> |**SVE-Math**|Qwen2.5-7B|51.3|
> |SVE-Math(-)|DeepSeek-7B|42.3|
> |**SVE-Math**|DeepSeek-7B|48.7|
> # 5. How much more computational cost and inference time is introduced by GeoGLIP?
> SVE-Math-7B introduces minimal computational overhead, as detailed in the below comparison table. The GeoGLIP encoder and Connector contribute an additional parameter size of 32.65MB and 8.73MB, and the Projectors accounting for 16.13MB. The inference time per sample increases slightly, from 19.80s to 20.04s (+0.24s). Training is conducted on 8 A100 GPUs with a batch size of 128 using the MathV360K dataset, which includes 40K images and 360K question-answer pairs. The total training time shows only a marginal increase, from 10.35h to 10.54h (+0.19h), demonstrating scalability for larger models and datasets.
>
> |Model|GeoGLIP|Connctor|Projectors|Time (inference/per sample)|Time (training/MathV360K)|
> |:-:|:-:|:-:|:-:|:-:|:-:|
> |G-LLaVA|-|-|16.52MB|19.80s|10.35h|
> |**SVE-Math**|32.65MB|8.73MB|31.20MB|20.04s|10.54h|

---

> ### Author Response · Authors · 2024-12-02
>
> We would like to sincerely thank you for your valuable feedback and thoughtful suggestions on our paper. We have carefully considered your recommendations, addressed your concerns, and updated both the revised paper and our responses accordingly.
>
> If you have any further questions or concerns, we would be delighted to address them promptly. Your insights are crucial to us, and we deeply appreciate the time and effort you have dedicated to reviewing our work.
>
> To summarize our main contributions:
>
> We systematically identify and analyze the impact of visual recognition errors on the mathematical reasoning performance of MLLMs, highlighting the critical role of accurately perceiving geometric primitives. This new aspect is orthogonal to existing methods focused on improving reasoning.
>
> We designed GeoGLIP, a lightweight, geometry-aware visual model with multitask learning capabilities, including shape grounding, junction detection, and boundary detection. GeoGLIP integrates seamlessly with diverse LLM backbones without requiring modifications to their reasoning components. Despite adding less than a 50MB increase in parameter size and only a 0.24s increase in inference time per image, and without relying on additional mathematical instruction datasets, our approach achieves an 8–12\% improvement in top-1 accuracy compared to the baseline (using LLaMA2-7B as the base LLM).
>
> When paired with advanced LLMs like DeepSeek and Qwen, our 7B model achieves performance comparable to GPT-4V, with 51.3\% and 48.7\% on the challenging MathVista benchmark, versus 49.9\% for GPT-4V.  While our 7B model does not surpass state-of-the-art MLLMs with 40B/70B parameters achieving over 60\% accuracy, integrating GeoGLIP into such large-scale LLMs is currently computationally prohibitive due to our limited resources.   We hope this work inspires further research into more effective fine-grained visual understanding in MLLMs. To support the community and assist other researchers in scaling our method to larger models and datasets,
> we will release the model weights, training scripts, and inference codes to facilitate broader adoption and experimentation.

---

### Author Response · Authors · 2024-11-28

## We sincerely thank the reviewers for their insightful feedback and constructive suggestions. We are delighted that our approach, SVE-Math, and the GeoGLIP module have been recognized as innovative and logical steps toward addressing the limitations of current multimodal large language models (MLLMs) in visual mathematical reasoning.
\
\
We have addressed all comments in individual responses to each reviewer.
\
\
\
Below, we address the key points raised across the reviews and clarify several aspects of our work to better demonstrate its contributions and implications.
## 1. Clarification of Goal and Contribution
* Our primary goal is not to solve the entire spectrum of mathematical reasoning tasks but to enhance the visual grounding capabilities of MLLMs in a way that complements their reasoning abilities. Our approach, which integrates a geometry-rich visual encoder (GeoGLIP), is orthogonal to existing methods focused on improving reasoning. By doing so, we aim to address the persistent bottleneck of fine-grained visual perception in mathematical contexts, as detailed in Section 1 and supported by the systematic analysis in Figure 1 and Figure 5a of the paper.

* GeoGLIP serves as a lightweight, domain-specific enhancement, specifically addressing geometric visual recognition errors. Importantly, it is designed to work seamlessly with diverse LLM backbones without requiring modifications to their reasoning components. This adaptability underscores its novelty and broad applicability.

##  2. Effectiveness of GeoGLIP and Dataset Independence
*  A key concern raised by reviewers is whether the improvement observed with our approach stems primarily from the instruction dataset used (Geo170K). To address this, we emphasize that the comparison with G-LLaVA [Gao et al., 2023a] in our paper is conducted under controlled conditions. Both G-LLaVA and our model use the same LLM backbone (LLaMA2-7B) and the Geo170K dataset, ensuring that any performance differences arise from the inclusion of the GeoGLIP encoder rather than the instruction dataset. The comparison results of Tables 1-3 confirm the effectiveness of our GeoGLIP, which significantly enhances visual mathematical reasoning performance. As shown in Tables 1 and 2, G-LLaVA with GeoGLIP (our SVE-Math-7B) improves Top-1 accuracy by 7.7\% on MathVerse and 12.3\% on MathVista. The improvement is not trivial, and our SVE-Math-7B achieves comparable performance to Math-LLaVA-13B on MathVerse (19.0\% vs. 21.2\%),  particularly excelling in the 'visual-only' scenario (16.4\% vs. 20.3\%). This scenario strips away the entire textual input, conveying the problem solely through the diagram.

* Additionally, we conducted further experiments integrating GeoGLIP with various LLM backbones that exhibit stronger mathematical reasoning abilities compared with LLaMA2-7B (e.g., Qwen2.5-Math-7B-Instruct and DeepSeek-Math-7B-Instruct, two of the most advanced mathematical reasoning LLMs currently available).  We evaluate our model on the most challenging MathVista benchmark, achieving 51.3\% and 48.7\% Top-1 accuracy, compatible with GPT-4V's performance (49.9\%).  Again, we observe a consistent 6\%-7\% improvement compared to the variant excluding GeoGLIP featurs as additional visual prompts.  These results reaffirm the complementary nature of the GeoGLIP visual encoder with reasoning abilities and highlight its generalizability benefits across diverse architectures. We will release those model weights, the training, and the inference codes to facilitate the computer vision community.

* Finally, we would like to clarify the distinction between our synthetic math-specific datasets and traditional mathematical instruction datasets. We do not create or use any additional self-generated instruction datasets beyond the publicly available Geo170K and MathV360K datasets for MLLM training. Instead, our synthetic samples, annotated with box/pixel-level details, are exclusively utilized to train the GeoGLIP visual encoder.  Compared to constructing mathematical instruction datasets, our synthetic data generation process is significantly more efficient and resource-friendly. It does not require manual labeling, as all data can be programmatically generated, e.g., through the Matplotlib Python library. In contrast, constructing instruction datasets often relies on GPT-4o to create diverse prompts and necessitates human intervention, making the process labor-intensive and costly. Moreover, training the lightweight, visual-centric GeoGLIP involves straightforward training recipes. In comparison, instruction tuning for MLLMs requires intricate configurations, such as carefully curated batch sizes and learning rates, as noted in [Shengbang et al., 2024].
> Shengbang Tong et al., Cambrian-1: A Fully Open, Vision-Centric Exploration of Multimodal LLMs, arXiv 2024.

---

> ### Author Response · Authors · 2024-11-28
>
> ## 3. Performance and Efficiency
> As detailed in Tables 1–4, SVE-Math achieves substantial improvements on benchmarks such as MathVerse and MathVista and GeoQA.  Notably, when based on LLaMA-series LLMs, it outperforms all models with the same configurations and is even comparable to larger-scale models like Math-LLaVA-13B, while maintaining computational efficiency. Equipping SVE-Math with more advanced LLMs significantly boosts performance. We recognize the importance of computational efficiency. SVE-Math with lightweight GeoGLIP and Connector introduces only a minimal computational overhead, with less than 50MB\% increase in parameter size and 0.24s\% increase in inference time per image, ensuring scalability to larger models and datasets. Detailed efficiency metrics will be added to the revised manuscript.
> ## 4. Ablation Studies and Model Outputs
> The paper already includes ablations to validate GeoGLIP (G-LLaVa vs. SVE-Math-7B in Tables 1–3) and the effect of individual visual features from GeoGLIP (middle panel of Table 5a). To address the reviewers' concerns, we have conducted additional experiments, including testing SVE-Math with different LLM backbones (e.g., Qwen2.5-Math-7B-Instruct and DeepSeek-Math-7B-Instruct), evaluating the impact of math-specific fine-tuning, and replacing the vision encoder with alternative models. These results consistently demonstrate the value of GeoGLIP in improving visual mathematical reasoning. We will also include visualizations of model outputs, synthetic data generation details, and additional qualitative results in the revised paper to strengthen the presentation.
>
> ## 5. Revisions
> In response to reviewers' suggestions, we will reorganize the paper to allocate more space for synthetic data descriptions, model outputs, and additional ablation results. The Methodology section will be streamlined by moving training details to the appendix, ensuring clarity and balance. Note that we obtained the Qwen2.5 implementation results after the revision submission deadline. These will be included in the final version.
>
> \
> \
> **We truly hope these clarifications and additional experiments address the reviewers' concerns and showcase the merit of our work.**
>
> \
> Kind regards,
> \
> Authors

---

### Author Response · Authors · 2024-11-29
**Explanation for Delayed Response**

Esteem AC and reviewers,

Thank you for your thoughtful review and feedback. We sincerely apologize for the delay in providing our responses, and we are truly sorry it took us so long to post a rebuttal. The delay was due to the time required to conduct additional experiments and obtain the necessary results to thoroughly address the concerns raised (7 days+28 hours +2 days and 18 hours +26 hours +12.7 hours+2 days). Once the results were available, we promptly submitted the rebuttal and paper revision to ensure the response was both accurate and comprehensive.

* Experiments on Cutting-edge LLMs (in response to all reviewers): Evaluating the effectiveness of our geometric-aware visual encoder (GeoGLIP) requires comprehensive ablation studies on state-of-the-art LLMs like DeepSeek and Qwen. Specifically, we compare GeoGLIP with configurations using only the CLIP visual encoder. These experiments involve four configurations, all trained on the MathV360K and Geo170K datasets.  We have access to only one machine with 8 GPUs, which limits our ability to run experiments in parallel. The entire training process takes approximately **one week** to complete.

* Experiments on other visual-centric models or training with different datasets (in response to reviewer yC4U): Utilizing 8 A100 GPUs, training another detector (Grounding DINO) on our math-specific datasets requires approximately **28 hours**, while training our visual model on an alternative dataset takes around **2 days and 18 hours**. Furthermore, integrating these visual features into the LLaVA-7B model on the Geo170K dataset takes an additional **26 hours** in total.

* Experiments on GLIP and directly providing geometric-relevant information  (in response to Reviewer M9gf):  Integrating the GLIP hierarchical pyramid features into MLLMs on the Geo170K dataset requires **12.7 hours**. As suggested by the reviewer, we explored directly providing geometric-relevant information to MLLMs. We could not find existing mathematical instruction datasets that include location information for geometric objects (e.g., bounding box coordinates or junction points). To address this, we inferred Geo170K training images using GeoGLIP to extract relevant location information, which we appended the special tokens <image> in `huam value`, using instructions such as: "there is a bounding box at ⟨x, y, w, h⟩". We need to controll the number of detected objects per image to ensure suitability (set to 10 objects). This entire data processing step, along with MLLM training, required approximately **2 days** to complete.

We deeply value the opportunity to engage with the review process and clarify any concerns. We hope the additional evidence and explanations provided in our rebuttal address the key points and demonstrate the significance of our work. We hope reviewers are still willing to give us a chance.

Thank you for your understanding, and we appreciate your time and consideration.

\
Kind regards,
\
Authors

---

### Author Response · Authors · 2024-12-02

Esteem AC and reviewers,

Thank you for your valuable feedback and thoughtful suggestions. We deeply appreciate the time and effort you have dedicated to reviewing our work.

In our previous response, we addressed concerns raised in the reviews and provided detailed explanations to clarify various aspects of our work. However, to ensure clarity and emphasize the core contributions of our work, we would like to briefly summarize the main points here:

* We systematically identify and analyze the impact of visual recognition errors on the mathematical reasoning performance of MLLMs, highlighting the critical role of accurately perceiving geometric primitives. This new aspect is orthogonal to existing methods focused on improving reasoning.

* We designed GeoGLIP, a lightweight, geometry-aware visual model with multitask learning capabilities, including shape grounding, junction detection, and boundary detection. GeoGLIP integrates seamlessly with diverse LLM backbones without requiring modifications to their reasoning components. Despite adding less than a 50MB increase in parameter size and only a 0.24s increase in inference time per image, and without relying on additional mathematical instruction datasets, our approach achieves an 8–12\% improvement in top-1 accuracy compared to the baseline (using LLaMA2-7B as the base LLM).

* We paired GeoGLIP with advanced LLMs like DeepSeek and Qwen and our 7B model achieves performance comparable to GPT-4V, with 51.3\% and 48.7\% on the challenging MathVista benchmark, versus 49.9\% for GPT-4V.  While our 7B model does not surpass state-of-the-art MLLMs with 40B/70B parameters achieving over 60\% accuracy, integrating GeoGLIP into such large-scale LLMs is currently computationally prohibitive due to our limited resources.

* We hope this work inspires further research into more effective fine-grained visual understanding in MLLMs. To support the community and assist other researchers in scaling our method to larger models and datasets, we will release the model weights, training scripts, and inference codes to facilitate broader adoption and experimentation.

\
Kind regards,
\
Authors

---

### Note · Authors · 2025-01-23

I have read and agree with the venue's withdrawal policy on behalf of myself and my co-authors.